# Chemogenomic profiling of breast cancer patient-derived xenografts reveals targetable vulnerabilities for difficult-to-treat tumors

Paul Savage [1,2], Alain Pacis[1,3], Hellen Kuasne[1], Leah Liu[1], Daniel Lai [4], Adrian Wan[4], Matthew Dankner[1,2], Constanza Martinez[1,5], Valentina Muñoz-Ramos [1], Virginie Pilon[1], Anie Monast[1], Hong Zhao[1], Margarita Souleimanova[1], Matthew G. Annis [1], Adriana Aguilar-Mahecha[6], Josiane Lafleur[6], Nicholas R. Bertos[1], Jamil Asselah[7], Nathaniel Bouganim[7], Kevin Petrecca[8], Peter M. Siegel [1,2], Atilla Omeroglu[5], Sohrab P. Shah [4,9], Samuel Aparicio [4], Mark Basik[6,10], Sarkis Meterissian[11] & Morag Park [1,2,5,12 ✉]

Subsets of breast tumors present major clinical challenges, including triple-negative, metastatic/recurrent disease and rare histologies. Here, we developed 37 patient-derived xenografts (PDX) from these difficult-to-treat cancers to interrogate their molecular composition and functional biology. Whole-genome and transcriptome sequencing and reverse-phase protein arrays revealed that PDXs conserve the molecular landscape of their corresponding patient tumors. Metastatic potential varied between PDXs, where low-penetrance lung micrometastases were most common, though a subset of models displayed high rates of dissemination in organotropic or diffuse patterns consistent with what was observed clinically. Chemosensitivity profiling was performed in vivo with standard-of-care agents, where multi-drug chemoresistance was retained upon xenotransplantation. Consolidating chemogenomic data identified actionable features in the majority of PDXs, and marked regressions were observed in a subset that was evaluated in vivo. Together, this clinically-annotated PDX library with comprehensive molecular and phenotypic profiling serves as a resource for preclinical studies on difficult-to-treat breast tumors.

---

[1] Rosalind & Morris Goodman Cancer Research Centre, McGill University, Montréal, QC H3A 1A3, Canada. [2] Department of Medicine, McGill University, Montréal, QC H4A 3J1, Canada. [3] Canadian Centre for Computational Genomics, McGill University and Genome Quebec Innovation Centre, Montréal, QC H3A 0G1, Canada. [4] Department of Molecular Oncology, British Columbia Cancer Research Centre, University of British Columbia, Vancouver, BC V5Z 1L3, Canada. [5] Department of Pathology, McGill University, Montréal, QC H4A 3J1, Canada. [6] Lady Davis Research Institute, Jewish General Hospital, Montréal, QC H3T 1E2, Canada. [7] Department of Oncology, McGill University, Montréal, QC H4A 3T2, Canada. [8] Department of Neurology and Neurosurgery, McGill University, Montréal, QC H3A 2B4, Canada. [9] Computational Oncology, Memorial Sloan Kettering Cancer Center, New York, NY 10065, USA. [10] Department of Surgery, Jewish General Hospital, Montréal, QC H3T 1E2, Canada. [11] Department of Surgery, McGill University Health Centre, Montréal, QC H4A 3J1, Canada. [12] Department of Biochemistry, McGill University, Montréal, QC H3A 1A3, Canada. ✉email: morag.park@mcgill.ca

Breast cancer comprises a heterogeneous collection of malignancies exhibiting distinct disease trajectories[1]. The histological and molecular diversity between tumors have been associated with important clinical phenotypes, namely metastatic potential and therapeutic response, which dictate survival[2,3]. Certain subsets of breast cancer patients represent unique clinical challenges, including triple-negative breast cancers (TNBC), which currently lack targeted therapies, metastatic disease, which is broadly treatment-resistant, and rare histological variants, where evidence-based guidelines are deficient[4,5].

Translational research relies on preclinical models as approximations of human tumors to address these challenges. Although cell lines and genetically engineered mouse models have contributed to seminal advances in our understanding of breast cancer biology, they have largely failed to account for inter- and intra-tumor heterogeneity, at least partially contributing to the high rate of attrition in oncologic drug development[6-9]. Due to the human-origin and limited selective pressures of immediate transplantation, patient-derived xenografts (PDX) have emerged as models for preclinical drug testing[6,10,11]. Although previous efforts to develop and characterize breast cancer PDXs have demonstrated the relative molecular fidelity of these models, their ability to recapitulate clinical phenotypes is of equal importance, yet remains unclear[12-16]. Anecdotal evidence supports the retention of therapeutic response and metastatic propensity upon xenotransplantation, but this has still not been systematically evaluated across PDX libraries with extensive clinical and molecular annotation[12,13,17].

Here we develop a series of PDXs representing breast tumors with unmet clinical needs. Molecular characterization is performed by whole-genome (WGS) and transcriptome (RNA-seq) sequencing and reverse-phase protein array (RPPA), which is complemented by in vivo evaluation of metastatic dissemination and chemosensitivity. We demonstrate that these models recapitulate the biology of parental human tumors and that the PDX platform serves as a tool for discovery and testing of precision therapeutics for those tumors showing poor responses to conventional chemotherapeutics.

## Results

### Establishment of poor prognosis breast cancer PDX library.
To develop preclinical models of breast cancers with unmet clinical needs, patient-derived xenografting of select cases was integrated into an existing biobanking protocol with extensive clinicopathological annotation to allow for molecular and functional interrogation (Fig. 1a). The selection criteria included tumors that were: (1) ER−; (2) HER2+; (3) high-grade ER+; (4) metastatic; or (5) rare histological variants. Orthotopic engraftment into female immunocompromised mice was successful for 36 tumor samples of 81 attempts, for an overall take rate of 44.4% (Fig. 1b). Engraftment success associated with the following clinicopathologic features: high-grade, low-ER expression (≤15%), HER2-negativity, germline BRCA1/2 mutation, previous systemic treatment and presence of axillary lymph node (ALN) metastases (Supplementary Fig. 1a). Supporting the aggressive biology of the cohort, successful engraftment was significantly associated with shorter progression-free survival (PFS) among patients whose tumors were used to attempt xenotransplantation (Log-rank $p = 0.027$) (Fig. 1c). Clinicopathologic features of the patients and PDXs are shown in Table 1, which demonstrates the high representation of TNBCs (73.0%) and rare histological variants (18.9%) in the PDX cohort.

Altogether, the Goodman Cancer Research Centre (GCRC) PDX library represents an aggressive breast cancer cohort comprised of 37 novel PDX lines derived from 36 tumors from 34 unique patients. One patient had three PDX models derived from their tumors—two sublines from distinct histological regions from their primary tumor (GCRC1784Xd/c, discussed below) and one from mediastinal lymph node metastasis (GCRC2054X) that developed at a later time point. Another patient had two PDXs developed from their tumors—one from their primary tumor (GCRC1915X) and another from a lung metastasis (GCRC2076X), which was sampled at the time of recurrence.

Tumor growth kinetics were evaluated over serial passages, where the median time to endpoint (10 mm in largest diameter) on first transplant generation was 128 days (range 30–234 days), and significantly decreased over subsequent passages ($p < 0.001$) (Fig. 1d, Supplementary Fig. 1b). Unlike engraftment rates, the only clinicopathologic parameter that was significantly associated with growth kinetics was previous exposure to therapy, where pre-treated PDX lines grew faster ($p = 0.004$) (Supplementary Fig. 1c). To address the feasibility of prospective drug testing using PDXs, the time to endpoint of passage two (P2) (a timeframe conducive for in vivo drug sensitivity studies) was compared to time-to-progression. Only 12.5% (3/24) of primary tumors reached P2 endpoint prior to patient progression; however, 63.6% (7/11) of patients with recurrent or metastatic disease progressed within this timeframe (Fig. 1d). This highlights a potential barrier for prospective personalized drug sensitivity testing using PDXs as avatars in advanced breast cancer. Together, the engraftment success rates, association with clinicopathologic factors and outcomes as well as growth kinetics described herein are consistent with previously published breast PDX cohorts[12,13,15].

### PDXs retain histopathological features of primary tumor.
To assess the histopathological fidelity of PDXs, we evaluated patient tumor–PDX pairs for concordance using a panel of immunohistochemical (IHC) markers. Due to the potential for lymphoproliferative outgrowths arising during xenotransplantation, all xenografts were initially screened for epithelial (pan-cytokeratin, pan-KRT) and lymphoid (CD45) markers to validate the epithelial origin of each PDX line (Supplementary Fig. 2a)[18]. Two lines were found to contain primary xenografts that were CD45+ (GCRC1924 and GCRC2034), both of which were derived from tumors with florid lymphocytic infiltrate (Supplementary Fig. 2b). Although one of the four GCRC1924 primary outgrowths established as a pan-KRT+/CD45− tumor and could be serially propagated as such, all P1 transplants of GCRC2034 were pan-KRT−/CD45+ and therefore this line was excluded from the final series.

Further evaluation of breast cancer-associated markers (ER, HER2, Ki67, p53, vimentin, CK5/6, CK8/18) revealed IHC diversity across the PDX library, with striking similarities between parental tumors and PDXs (Fig. 1e, Supplementary Fig. 3). Regarding receptor expression, ER and HER status were concordant in 80.6% and 100%, respectively (Fig. 1f). Of the seven cases with ER discordance, five were from patients whose tumors were low-ER-expressing (1–15%) and corresponding PDXs were ER negative (GCRC1715, 1784, 1882, 2001, 2047). Conversely, re-expression of ER in the PDX occurred in two lines. One was an ER− skin recurrence from a patient who was initially diagnosed with an ER+ primary breast cancer and was sampled for xenografting while the patient was being treated with a selective-estrogen receptor degrader (GCRC1971). The other was an ER− brain metastasis from a patient who initially had an ER+ primary breast cancer (GCRC1944) (Fig. 1e, f; primary tumors not shown).

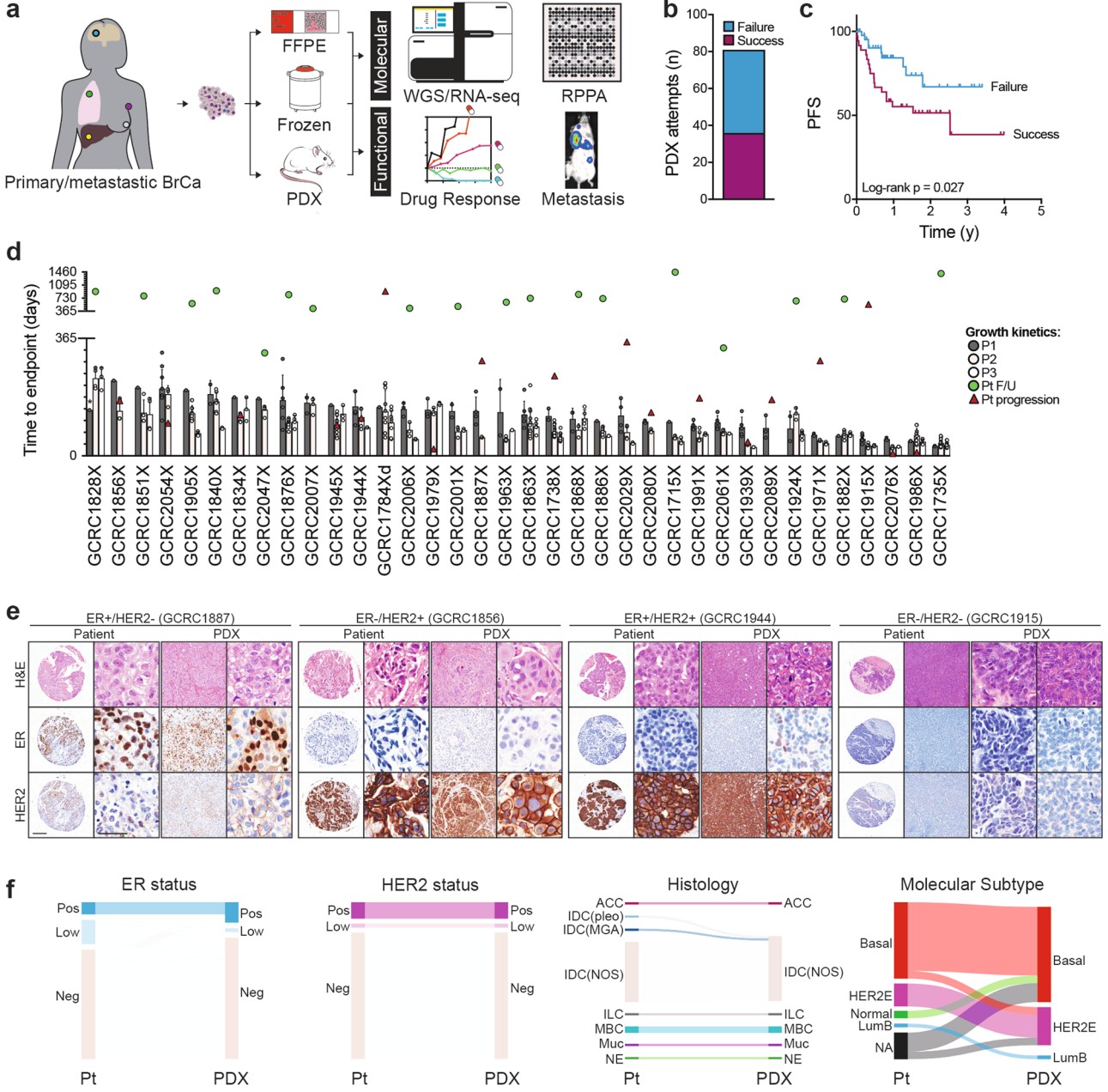

**Fig. 1 Breast cancer PDX collection. a** Schematic of live biobanking protocol of primary and metastatic breast cancer (BrCa). Fresh breast tumors were divided for traditional biobanking (FFPE and snap frozen) and generation of PDX. These samples underwent molecular (WGS/RNA-seq and RPPA) and functional (in vivo drug sensitivity, metastasis assays) analyses. FFPE formalin-fixed paraffin-embedded, PDX patient-derived xenograft, WGS whole-genome sequencing, RNA-seq RNA-sequencing, RPPA reverse-phase protein array. **b** Bar chart of success rate for engraftment attempts per tumor sample. Success (n = 36), failure (n = 45). **c** Kaplan–Meier analysis of progression-free survival (PFS) for patients whose tumors successfully generated PDXs versus those that failed. **d** Bar chart of growth kinetics (time to endpoint, defined as growth to 10 mm) across passage 1–3 (P1, P2, P3) of PDX models (P1 median n = 2.5, range 1–11; P2 median n = 6, range 3–29; P3 median n = 4, range 3–15). Mean ± SD. Patient follow-up (Pt F/U) time and time to progression (Pt progression) are overlaid. *Mouse sacrificed before endpoint reached. **e** Representative H&E, estrogen receptor (ER) and HER2 staining from patient tumor and corresponding PDXs, representing the four major clinical subtypes. Left, whole-image scale bar 200 μm; right higher-magnification scale bar 50 μm. Complete staining panel in Supplementary Fig. 3. **f** Sankey diagrams for patient (Pt) and PDX concordance for estrogen receptor (ER) status, HER2 status, histological subtype and gene expression subtype (AIMS, absolute intrinsic molecular subtype) (left to right). ACC adenoid cystic carcinoma, IDC (pleo) invasive ductal carcinoma (pleomorphic), IDC (MGA) IDC in microglandular adenosis background, ILC invasive lobular carcinoma, MBC metaplastic breast cancer, Muc mucinous, NE neuroendocrine, BL basal-like, HER2E HER2-enriched, NL normal-like, LumB luminal B.

Architectural features and lineage markers were most notable among rare histological variants. This includes the cylindromatous organization with p63+ basal cells from an adenoid cystic carcinoma (ACC, GCRC1828); synaptophysin expression in a neuroendocrine carcinoma (NE, GCRC1979); mucicarmine+ lakes from a mucinous tumor (GCRC2007); loss of E-cadherin in an invasive lobular carcinoma (ILC, GCRC1971); and vimentin+ mesenchymal-like cells dispersed in a chondroid matrix in a metaplastic breast cancer (GCRC1784) (Fig. 1f, Supplementary Fig. 4a). In the latter example, both ductal and

**Table 1 Clinicopathologic characteristics of patient tumors and PDXs.**

| Patient | | | | | | | | PDX | | | | | | |
|---|---|---|---|---|---|---|---|---|---|---|---|---|---|---|
| Tumor ID | Site | Histology | ER | HER2 | Subtype | gBRCA1/2 | Therapy[a] | ALN | Distant | PDX ID | Histology | ER | HER2 | Subtype |
| GCRC1715T | Breast | IDC-NOS | 1% | − | HER2E | | A, C, T | + | − | GCRC1715X | IDC-NOS | − | − | HER2E |
| GCRC1735T | Breast | IDC-NOS | − | FISH 2.2 | Basal | BRCA1 | A, C, T, H | − | − | GCRC1735X | IDC-NOS | − | FISH 2.2 | Basal |
| GCRC1738T | Breast | IDC-NOS | − | − | Basal | BRCA1 benign | A, C, T | + | + | GCRC1738X | IDC-NOS | − | − | Basal |
| GCRC1784T[b] | Breast | MBC (chondroid) | 15% | − | Basal | | A, C, T | + | + | GCRC1784Xd / GCRC1784Xc | MBC (chondroid) / IDC-NOS | − | − | Basal / Basal |
| GCRC2054T[b] | MLN | Met Ca (1° MBC chondroid) | − | − | N/A | | A*, C*, T*, M*, F* | + | + | GCRC2054X | IDC-NOS | − | − | Basal |
| GCRC1828T | Breast | ACC | − | − | Normal | | None | + | + | GCRC1828X | ACC | − | − | Basal |
| GCRC1834T | Breast | IDC-NOS | − | − | Basal | | None | + | + | GCRC1834X | IDC-NOS | − | − | Basal |
| GCRC1840T | Breast | IDC-NOS | − | − | Basal | BRCA1 | None | + | − | GCRC1840X | IDC-NOS | − | − | Basal |
| GCRC1851T | Breast | IDC-NOS | − | − | Basal | | None | − | + | GCRC1851X | IDC-NOS | − | − | Basal |
| GCRC1856T | Brain | Met Ca (1° IDC-NOS) | − | + | HER2E | | H, V | − | + | GCRC1856X | IDC-NOS | + | + | HER2E |
| GCRC1863T | Breast | IDC in MGA | 10% | − | Basal | | E, C, T | − | − | GCRC1863X | IDC-NOS | 10% | − | Basal |
| GCRC1868T | Breast | IDC-NOS | − | − | Basal | | T, nab-T | − | − | GCRC1868X | IDC-NOS | − | − | Basal |
| GCRC1876T | Breast | IDC-NOS | − | − | Basal | | L | − | − | GCRC1876X | IDC-NOS | − | − | Basal |
| GCRC1882T | Breast | IDC-NOS | 5% | − | Basal | | None | − | − | GCRC1882X | IDC-NOS | − | − | Basal |
| GCRC1886T | Breast | IDC (pleomorphic) | − | − | Normal | | A, C, Car, T | + | + | GCRC1886X | IDC-NOS | + | − | Basal |
| GCRC1887T | Brain | Met Ca (1° MBC squamous) | + | − | HER2E | BRCA2 | A*, C, T*, L*, Ex*, Cap*, V*, D, Car | − | + | GCRC1887X | MBC (squamous) | + | − | HER2E |
| GCRC1905T | Breast | IDC-NOS | − | − | Basal | | None | − | + | GCRC1905X | IDC-NOS | − | − | Basal |
| GCRC1915T[c] | Breast | IDC-NOS | − | − | Basal | | A, C, Car, T | − | + | GCRC1915X | IDC-NOS | − | − | Basal |
| GCRC2076T[c] | Lung | Met Ca (1° IDC-NOS) | − | − | N/A | | A*, C*, T*, Car*, M*, F* | + | + | GCRC2076X | IDC-NOS | − | − | Basal |
| GCRC1924T | ALN | Met Ca (1° IDC-NOS) | − | − | N/A | BRCA1 VUS | D, C | + | + | GCRC1924X | IDC-NOS | − | − | Basal |
| GCRC1939T | Breast | IDC-NOS | − | − | Basal | | A, C, T, D | + | + | GCRC1939X | IDC-NOS | − | − | Basal |
| GCRC1944T | Brain | Met Ca (1° IDC-NOS) | − (1° +) | + | N/A | | T*, Car*, H*, A*, C*, V*, Cap*, T-DM1, Tam | + | + | GCRC1944X | IDC-NOS | + | + | HER2E |
| GCRC1945T | Brain | Met Ca (1° IDC-NOS) | − | − | Basal | BRCA1 | A*, C*, D*, T, Car, Cap* | + | + | GCRC1945X | IDC-NOS | − | − | Basal |
| GCRC1963T | Breast | IDC-NOS | − | − | Basal | | None | + | + | GCRC1963X | IDC-NOS | − | − | HER2E |
| GCRC1971T | Skin | Met Ca (1° ILC) | − (1° +) | − | HER2E | | L, D*, C*, Fulv, Tas | + | + | GCRC1971X | ILC | + | − | HER2E |
| GCRC1979T | 2° Breast | NE | − | − | N/A | | E, C, T | + | + | GCRC1979X | NE | − | − | HER2E |
| GCRC1986T | Liver | Met Ca (1° IDC-NOS) | + | − | N/A | | A, C, T, Cis, C, F | − | + | GCRC1986X | IDC-NOS | + | − | Basal |
| GCRC1991T | Breast | IDC-NOS | − | + | HER2E | | A*, C*, T, H, V, T-DM1, ONT-380 | − | + | GCRC1991X | IDC-NOS | − | + | HER2E |
| GCRC2001T | Breast | IDC-NOS | 1% | − | Basal | BRCA1 | None | + | − | GCRC2001X | IDC-NOS | − | − | HER2E |
| GCRC2006T | Breast | IDC-NOS | − | − | LumB | | None | + | − | GCRC2006X | IDC-NOS | − | − | Basal |
| GCRC2007T | 2° Breast | Mucinous | + | − | LumB | | None | − | − | GCRC2007X | Mucinous | + | − | LumB |
| GCRC2029T | Breast | MBC (squamous) | − | − | Basal | | A, C, T, Cis | + | − | GCRC2029X | MBC (squamous) | − | − | HER2E |
| GCRC2047T | Breast | IDC-NOS | 1% | − | Basal | BRCA1 | A, C, T | + | − | GCRC2047X | IDC-NOS | − | − | Basal |
| GCRC2061T | Breast | IDC-NOS | − | + | HER2E | | None | + | + | GCRC2061X | IDC-NOS | − | − | HER2E |
| GCRC2080T | Breast | IDC-NOS | − | + | HER2E | BRCA1 | A, C, T, V, H, Cis | + | − | GCRC2080X | IDC-NOS | − | + | HER2E |
| GCRC2089T | Breast | IDC-NOS | − | − | N/A | | A, C, Car, T | | | GCRC2089X | IDC-NOS | − | − | Basal |

ALN axillary lymph node, gBRCA1/2 germline BRCA1/2 mutation, IDC-NOS invasive ductal carcinoma, not otherwise specified, MBC metaplastic breast cancer, Met Ca metastatic carcinoma, MGA microglandular adenosis, MLN mediastinal lymph node, NE neuroendocrine.
[a]Prior to engraftment (*indicates drug exposure >1 year prior to sampling for PDX): A adriamycin/doxorubicin, C cyclophosphamide, Cap capecitabine, Car carboplatin, Cis cisplatin, D docetaxel, E epirubicin, Ex exemestane, F 5-fluorouracil, Fulv fulvestrant, H herceptin/trastuzumab, L letrozole, M methotrexate, nab-T nab-paclitaxel, T taxol/paclitaxel, T-DM1 trastuzumab-emtansine, Tam tamoxifen, Tas taselisib, V vinorelbine.
[b]GCRC1784/2054 derived from same patient.
[c]GCRC1915/2076 derived from same patient.

chondroid histological components were observed in the GCRC1784 patient tumor, which was recapitulated in one of the P1 mice (P1-8), whereas the other P1 PDX displayed pure ductal morphology (P1-7) (Supplementary Fig. 4b, c). Serial transplantation resulted in purification of each histological component, which remained stable over subsequent passages, and the resulting sublines were further characterized separately (GCRC1784Xd, ductal; and GCRC1784Xc, chondroid) (Supplementary Fig. 4c). Altogether, this data supports the preservation of histopathologic features upon xenotransplantation.

**PDXs preserve the mutational landscape**. To define the molecular landscape of the PDX cohort, comprehensive genomic profiling was undertaken. WGS was performed on early-passage PDXs ($n = 36$, P1–3), corresponding patient tumors ($n = 31$) and germlines where the somatic mutational burden was evaluated at multiple levels. The median whole-genome single-nucleotide variant (SNV) load was 10,773 (range 2103–68,363), consistent with previous breast cancer analyses (Fig. 2a)[15,19]. At the extremes of SNV load was an ACC (GCRC1828) demonstrating low SNV burden ($n = 2103$) that contrasted with a hypermutated ($n = 68,363$) ILC (GCRC1971). The subtype of base substitutions (e.g. C>A, C>G, C>T, T>A, T>C, T>G) were well-preserved in PDXs (Fig. 2a)[20]. Variability across models was also observed in small insertions and deletions (indels), with a median of 1268 per tumor (range 304–2841) (Fig. 2a). When only coding regions were considered, there was a median of 113 non-synonymous, truncating or splice-site mutations per tumor (range 20–662) (Fig. 2a).

The effect of xenotransplantation on subclonal SNVs was assessed by comparing variant allele frequencies (VAF) across 34 PDX–primary tumor pairs. VAF correlations were highly variable, with a median correlation coefficient of 0.45 (range −0.21–0.72) for all variants and 0.63 (range −0.02–0.85) for coding variants (Supplementary Fig. 5a). Although a large proportion of variants lied on the scatter plot diagonal, indicating neutral clonal dynamics, all models demonstrated variant clusters along the axes, supporting an element of clonal selection upon xenotransplantation. For example, GCRC1863 ($r = 0.72$, $p < 2.2 \times 10^{-16}$) and GCRC1915 ($r = 0.71$, $p < 2.2 \times 10^{-16}$) showed strong preservation of mutation prevalence, in contrast to the GCRC2001 model, which displayed marked shifts in VAFs ($r = -0.21$, $p = 1.21 \times 10^{-77}$).

Larger structural variants (SV), including duplications, insertions, deletions, inversions and translocations, were observed with median of 415 events per tumor (range 51–1253), with marked similarity in genomic structures between parental tumors and PDXs (Fig. 2a, Supplementary Fig. 5a). Copy number alterations (CNA) were also maintained after xenotransplantation, though gains and losses appeared more pronounced in the PDX because of in silico purification by filtering murine stromal reads (Fig. 2b). Several of the rare histological subtypes were noted to display quiet copy number profiles, including the ACC (GCRC1828), ILC (GCRC1971), NE carcinoma (GCRC1979) and mucinous (GCRC2007) models, consistent with previous observations in these tumor types[21,22]. Arm-length CNAs previously associated with breast cancer (gains in 1q, 3q, 8q, 10p, 12p, 20q; losses in 5q, 8p, 11q, 16q), as well as amplifications/deletions (amplification of *ERBB2*, *ZNF703/FGFR1*, *CCND1*, *MYC*, *EGFR*; deletion of *RB1*, *PTEN*, *CDKN2A/B*), were represented within the cohort and recapitulated in the PDXs (Fig. 2b, c, Supplementary Fig. 5a).

Examining a panel of genes, which have been causally implicated in breast tumorigenesis revealed a long-tail distribution, where the frequencies of alterations in the PDX cohort were comparable to human breast cancer datasets, particularly ER- tumors (Supplementary Fig. 6). Hotspot oncogenic drivers, truncating loss-of-function mutations and CNAs were identified in members of critical breast cancer pathways, including P53, receptor tyrosine kinase (RTK), PI3K and MAPK signaling, as well as cell cycle, transcriptional and epigenetic regulators and were highly conserved in PDXs (Fig. 2c). Together, these data show that although there is variability in clonal evolution upon xenotransplantation at the genome-wide scale, PDXs faithfully maintain the diverse landscape of coding mutations and oncogenic drivers displayed in their parental breast tumors.

**PDXs preserve the expression landscape**. To evaluate the expression landscape, transcriptome sequencing was performed on 37 PDX samples and 29 matched patient tumors. Initial hierarchical clustering using the top 1000 most variable genes separated the PDXs from human tumors, driven by a gene set that is predominantly expressed in human samples (Supplementary Fig. 7a). Differential expression analysis between 29 human tumor–PDX pairs revealed 3037 genes displaying log2-fold higher expression in human samples, which were highly enriched for immune-related pathways (Supplementary Fig. 7b, c). After filtering these genes to perform an epithelial-centric analysis, all tumor–PDX pairs clustered adjacent to one another (Fig. 3a). Samples derived from an individual patient, including PDXs from metastatic lesions, clustered with one another (GCRC1784/2054 and GCRC1915/2076), further supporting retention of the gene expression landscape upon xenotransplantation.

Classification of samples by breast cancer intrinsic subtypes revealed 86.7% concordance for tumor–PDX pairs, where 26 (70.3%) of PDXs were basal-like, 10 (27.0%) were HER2-enriched (HER2E) and one (2.7%) was luminal B (Fig. 3a). Subtype switching from normal-like in human to basal-like in PDX was observed in tumors with high stromal content (GCRC1828 and GCRC1886), which was lost upon engraftment (Fig. 1f, Supplementary Fig. 3). The other two subtype switches were basal-like in human to HER2E in PDX (GCRC2001 and GCRC2029).

Pathway-level analysis was performed using single-sample gene set enrichment analysis (ssGSEA) across 2117 pathways ('C2 chemical and genetic perturbations' pathways from mSigDB) (Fig. 3b)[23]. Unbiased clustering of pathway activation appeared to be largely driven by subtype, where luminal, ERBB2 and basal expression modules were well-preserved (Fig. 3b, c). In addition to this, other pathways (e.g. cell cycle, interferon response, hypoxia) demonstrated variability within subtype. Despite distinct species microenvironments, significant correlations were observed for pathway activation scores between human tumor and PDXs for key pathways associated with breast tumorigenesis, including proliferation (Pearson $r = 0.66$, $p < 0.0001$), hypoxia (Pearson $r = 0.36$, $p = 0.03$) and EMT (Pearson $r = 0.47$, $p = 0.005$) (Fig. 3b, d).

To further interrogate the expression landscape at the protein and signaling levels, 37 PDX lines were subject to RPPA against total ($n = 176$), phosphorylated ($n = 64$), methylated ($n = 2$) and detyrosinated ($n = 1$) proteins. After removing an outlying sample known for high matrix protein production (GCRC1784 chondroid, correction factor 0.33), supervised differential expression analysis based on receptor status confirmed expression of proteins previously associated with ER (ER, AR, INPP4B), HER2 (pHER2) and TNBC (pATM, pChk2, Notch1, B-catenin, TP53, low PTEN) biology (Fig. 3e). At the pathway level, variability was seen in hormonal, cell cycle, apoptosis, DNA damage, EMT and

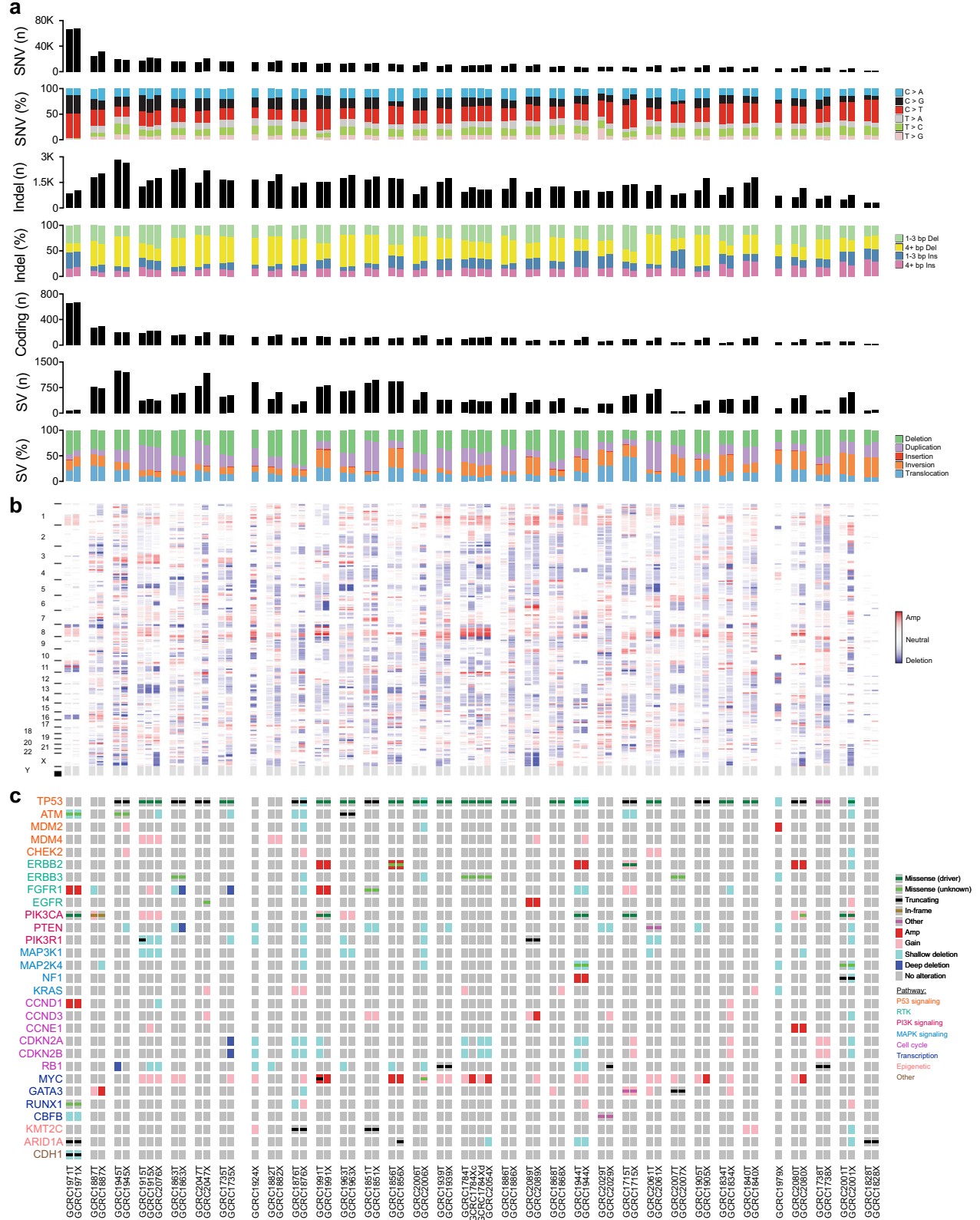

**Fig. 2 Genomic landscape of PDX library. a** Bar charts of somatic single-nucleotide variant (SNV) load and type, insertion/deletion (indel) load and type and structural variant (SV) load and type across PDX ($n = 36$) and patient tumor ($n = 31$) samples, which underwent whole-genome sequencing. Samples are ordered on SNV load and samples from same patient (patient tumor and PDX(s)) are adjacent to each other. **b** Heatmap of genome-wide copy number alteration (CNA) profiles. **c** Oncoprint of somatic alterations in genes previously associated with breast tumorigenesis. Sample order is the same for **a–c**.

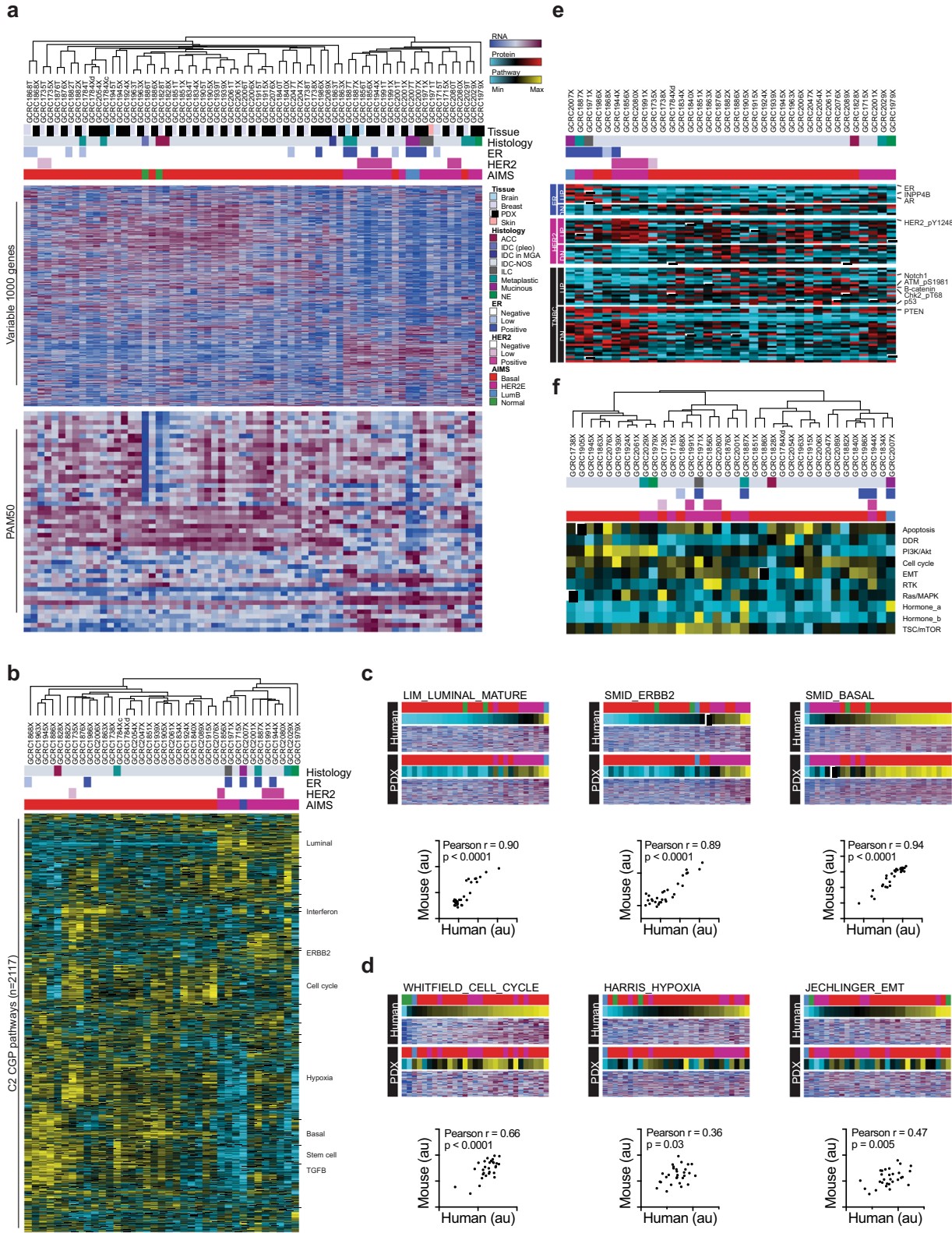

other specific signaling pathways (PI3K/Akt, RTK, Ras/MAPK, TSC/mTOR; Fig. 3f).

The overall correlation between RNA and RPPA data was high. At the gene level, 63.3% of transcripts/probes were significantly correlated across the PDX collection (B-H $p$-value < 0.05) (Supplementary Fig. 7d). This was consistent with TCGA data, where a significant correlation between RPPA/RNA-seq $R$ values

was observed in the 108 probes common to our and TCGA datasets (Pearson $r = 0.61$, $p < 0.0001$) (Supplementary Fig. 7e). Although several probes measuring post-translational modifications (PTM) were highly correlated with RNA levels (e.g. pHER2, p4E-BP1, pRB), there was an enrichment for PTM RPPA probes in those that did not correlate with RNA expression, further supporting the added-value of RPPA data (Fisher's exact test

**Fig. 3 Expression landscape of PDX library. a** Heatmap of unbiased hierarchical clustering across PDX ($n = 37$) and patient tumor ($n = 29$) samples, which underwent RNA-sequencing and filtering of stromal genes. Clustering based on top 1000 most variable genes (interquartile range), with histopathological (histology, ER, HER2) and subtype (absolute intrinsic molecular subtype, AIMS) annotation (top). Heatmap of PAM50 genes (below). **b** Heatmap of unbiased hierarchical clustering across PDX samples ($n = 37$) using ssGSEA scores for C2 CGP gene sets from MSigDB from RNA-seq data. **c** Heatmap of ssGSEA scores and gene expression (RNA-seq) for breast cancer subtype-associated pathways (luminal, ERBB2, basal). Patient tumor samples are ranked by increasing score (top) with corresponding PDXs (bottom) in the same order ($n = 29$). Correlation plot of ssGSEA pathway scores between human and PDX samples (below heatmap). The same applies for **d** for proliferation, hypoxia and epithelial-to-mesenchymal transition (EMT) signatures. **e** Heatmap of differentially expressed genes based on ER/HER2/TNBC status of PDX samples ($n = 36$), which underwent reverse-phase protein array (RPPA). **f** Heatmap of unbiased hierarchical clustering of RPPA pathway scores across PDX samples ($n = 36$). DDR DNA damage response, RTK receptor tyrosine kinase.

$p$-value < 0.0001). From the perspective of individual PDXs, 61.1% of models showed significant correlations across all transcripts/probes (B-H $p$-value < 0.05) (Supplementary Fig. 7f). Together, these data demonstrate that the PDX collection displays diverse expression programs that are representative of their parental tumors.

**PDXs model distinct patterns of metastatic dissemination.** As space-filling lesions in vital organs, metastatic disease is generally treatment-refractory and the primary cause of breast cancer mortality. To assess the metastatic potential of the breast PDXs, we screened 29 models by gross organ examination at the time of necropsy during routine xenograft passaging (Fig. 4a). Metastatic lesions were observed in 14 lines, the majority of which were micrometastases to the lung at <50% penetrance (Fig. 4a, b, Supplementary Fig. 8a).

In PDX models with the highest frequencies of metastasis, corresponding patients displayed significantly worse PFS (Log-rank $p = 0.004$) (Fig. 4c). Two of the highly metastatic PDX lines (GCRC1986 and 1991) appeared to be able to disseminate to multiple sites at high frequency, both of which were derived from patients who developed diffusely metastatic disease early in the course of their disease (Fig. 4a, d, h). To better quantify metastatic capacity, microscopic examination was performed on necropsy specimens in larger cohorts of mice ($n = 9$) after developing symptomatic metastases following primary resection when mammary fat pad tumor diameter reached 10 mm. The GCRC1986X model was generated from a liver metastasis in a patient initially diagnosed with locally advanced, ER+ breast cancer that disseminated to the liver and lungs within 10 months of diagnosis (Fig. 4d). At experimental endpoint (median 116 days, range 101–143), PDXs from this patient demonstrated high rates of metastasis to the ALN (66.7%), lung (88.9%), liver (88.9%), brain (55.6%) and within the abdomen (77.8%) (Fig. 4e–g). Similarly, the GCRC1991X model demonstrated broad metastatic propensity, and was developed from a patient who presented with synchronous metastatic HER2+ breast cancer (Fig. 4h). In the PDX, similarly high rates of metastasis were seen in the ALN (66.7%), lung (100%), liver (100%), brain (22.2%) and abdomen (55.6%) at experimental endpoint (median 101 days, range 81–136) (Fig. 4i–k).

Contrasting these widespread patterns of dissemination, the GCRC1971X PDX displayed organotropism. This ER+ ILC model was generated from a skin recurrence seven years after diagnosis (Fig. 4l). Following the development of severe twirling behavior in 2/4 (50%) mice from the first transplant generation of our metastasis screen, cranial dissection revealed the presence of large skull-base metastases (Fig. 4o). Clinical follow-up on this patient revealed a metastatic lesion to the cavernous sinus within one year of her local recurrence (Fig. 4p). A larger PDX cohort validated the initial observation, where all mice developed skull-base metastases with low frequency spread to other sites by endpoint (median 152 days, range 124–154) (Fig. 4m, n).

Together these data show that although the majority of PDXs display low-penetrance metastases to the lung and/or axilla, a subset can mimic the widespread or organotropic patterns of dissemination observed in the patient.

**PDXs retain the chemosensitivity profile of patients.** To systematically evaluate chemosensitivity profiles across the GCRC library, agents from the major classes of chemotherapeutics commonly used in advanced breast cancer were screened in a PDX clinical trial (PCT) using a "one mouse per treatment per model" approach (Supplementary Fig. 9a)[6]. The in vivo efficacy of doxorubicin, gemcitabine, cisplatin and paclitaxel were evaluated as single agents over a ~28-day study across 25–31 PDX lines. Responses were quantified and stringently categorized as complete response (CR), partial response (PR), stable disease (SD) or progressive disease (PD) according to the mRECIST criteria (Supplementary Table 1)[6]. By performing replicates across 13 different model-drug-response combinations, we found the PCT approach to be highly reproducible, where a high correlation between replicates was observed ($r = 0.9646$, $p < 0.0001$) (Supplementary Fig. 9b).

The fidelity of the PDX response was assessed as a clinical cohort and from the perspective of an individual patient. At the population level, variable response rates were observed for each drug compared to an untreated control group, as demonstrated by rightward shifts in the waterfall plots (Fig. 5a). The objective response rates (CR + PR) for each agent ranged from 8.0 to 51.5%, in line with those observed in single-agent studies in patients with metastatic breast cancers (Fig. 5b)[24–26]. These response rates translated to significant improvements in tumor volume doubling-free survival for each of the chemotherapy-treated cohorts compared to untreated mice (Log-rank $p < 0.005$ for each treatment arm) (Fig. 5c).

Diverse response profiles were observed by evaluating the chemosensitivity profiles of 30 PDXs tested with multiple single-agent regimens (Fig. 5d). Overall, the average response across the four chemotherapies evaluated in the PDXs correlated with the number of drugs the patient had been exposed to prior to engraftment (Fig. 5e). This multi-agent analysis could be used to further classify PDXs as broadly chemosensitive (≥2 CR/PR) or resistant (≤1 CR/PR). This classification based on PDX response was reflected in patient clinical outcomes, where patients with chemoresistant PDXs displayed worse PFS (Fig. 5f).

To evaluate whether individual tumors retain therapeutic sensitivity profiles following xenotransplantation, PDX response data were retrospectively compared with responses observed in the patient prior to engraftment. Sufficient patient data were available for doxorubicin, cisplatin and paclitaxel (no patients were treated with gemcitabine prior to engraftment) and patient responses were classified as sensitive (CR/PR) or resistant (SD/PD). PDXs from patients' whose tumors were clinically chemoresistant consistently exhibited decreased sensitivity compared to tumors that were sensitive and/or unexposed to the

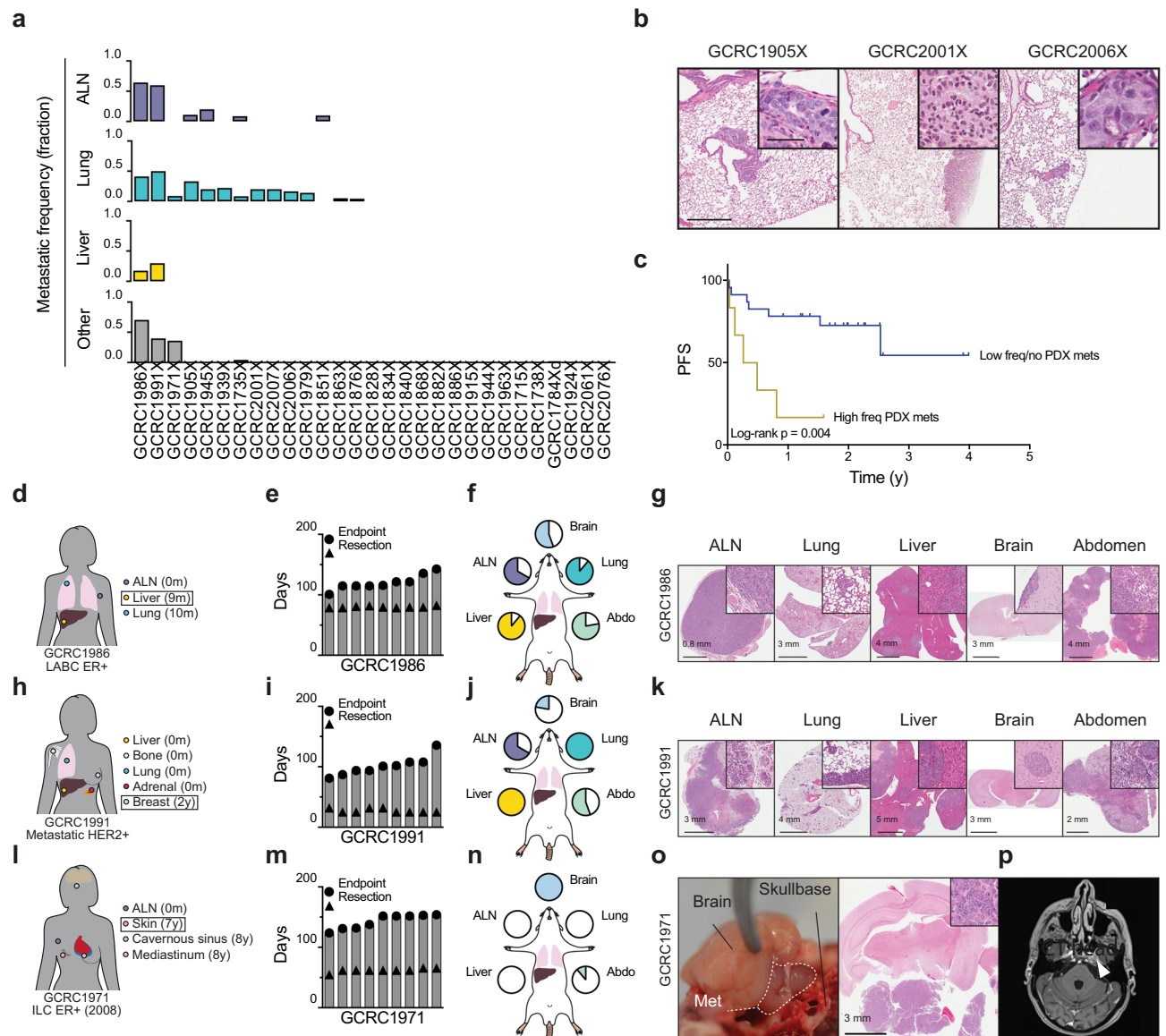

**Fig. 4 Metastatic potential of breast PDXs. a** Bar plots of fraction of animals with gross metastatic lesions at various anatomical sites at routine necropsy (median $n = 11$, range $n = 5$–$47$). ALN axillary lymph node. **b** Representative H&E images of lung metastatic lesions. Scale bar 400 μm, inset scale bar 50 μm. Complete lung micrometastasis H&E images are in Supplementary Fig. 7a. **c** Kaplan–Meier of progression-free survival (PFS) for patients whose PDXs developed frequent metastases (top 20%) in vivo versus those exhibiting less frequent or no metastases. **d** Schematic of metastatic lesions and timing since diagnosis for GCRC1986 patient. Squared sample is the lesion used to generate the PDX. **e** Bar chart of time of resection (at 10 mm) and endpoint for each PDX animal ($n = 9$). Bars represent individual mice. **f** Piecharts for frequency of metastasis observed to ALN, lung, liver, abdomen and brain. **g** Representative H&E images of metastatic lesions to specific anatomic sites. The same formatting applies to **h–k** for GCRC1991 and **l–n** for GCRC1971. **o** Gross picture of skull-base metastasis in PDX (left), with corresponding H&E image (middle). **p** Gadolinium MRI of head of the patient GCRC1971 shows 1.1 cm cavernous sinus mass (arrowhead).

given agent (Fig. 5g). Together, of the 34 patient-PDX–drug combinations that could be compared, 64.7% showed concordant responses (Fig. 5h). Half of the discordances were cases in which the patient showed doxorubicin sensitivity but was resistant in the PDX, which may be attributed to dose-limiting toxicity observed in the mouse when doxorubicin was administered above 3 mg/kg. Alternatively, this could be explained by engraftment of an enriched subpopulation of drug resistant subclones from residual disease following a partial response to neoadjuvant chemotherapy[27]. The mechanism by which three cases went from paclitaxel resistant in the patient to sensitive in the PDX remains unclear, though the effect of drug holiday during PDX generation resulting in re-sensitization may play a role[28,29].

**PDXs identify therapeutics for difficult-to-treat tumors**. To evaluate the utility of our chemogenomically profiled PDX library in the identification of therapies for difficult-to-treat tumors, actionability was assessed across WGS, RNA-seq, RPPA and chemosensitivity studies using previously published actionable features (OncoKB, ESMO/ESCAT, DEPO, Akbani et al.) (Fig. 6a)[30–33]. Potentially actionable alterations were observed in the overwhelming majority of models, including those with clinical ER/HER2/BRCA alterations (38.8%), as well as those without clinically used predictive biomarkers. DNA-level candidates supported by compelling clinical evidence included *PIK3CA* hotspot mutations (H1047R in GCRC1715X, GCRC1944X, GCRC1991X, GCRC2001X and GCRC2029X;

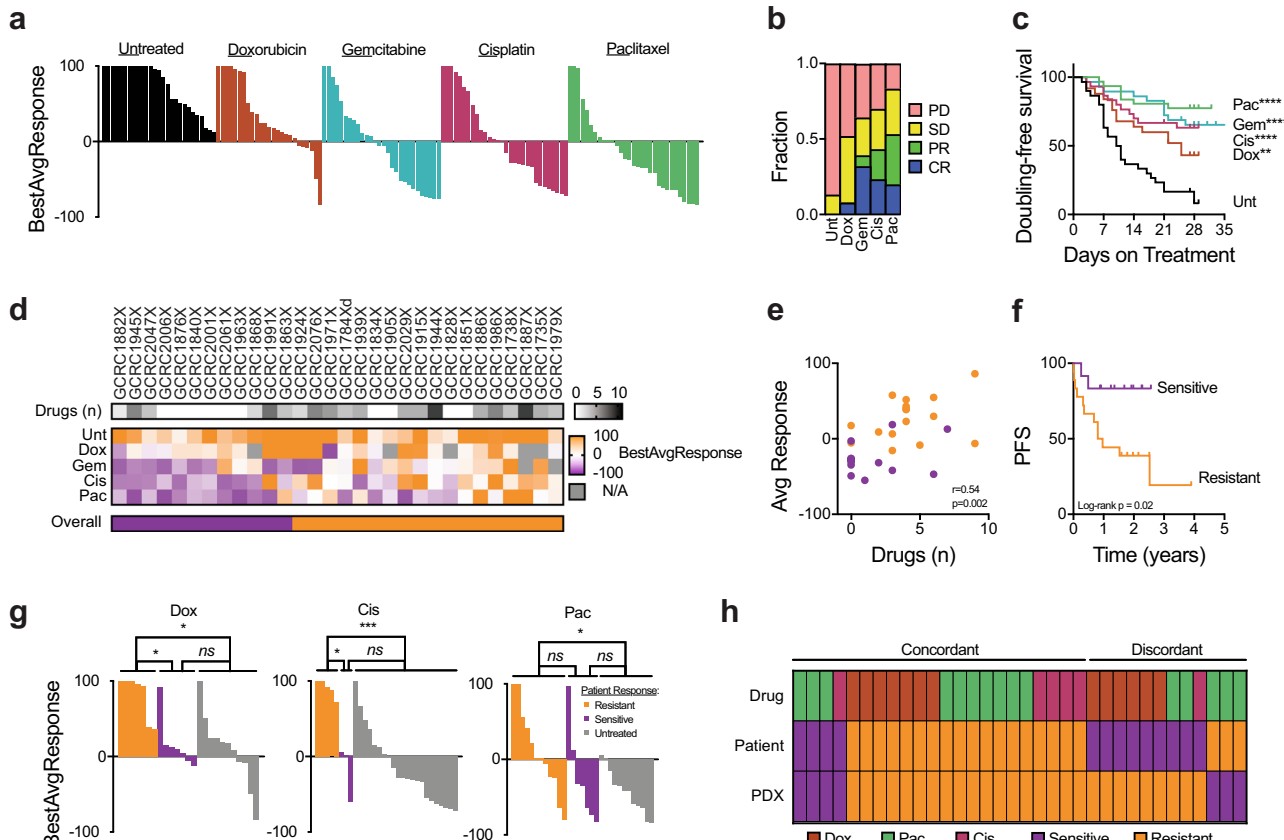

**Fig. 5 Therapeutic response of breast PDXs to standard-of-care chemotherapeutics. a** Waterfall plots for BestAvgResponse after ~28 days on treatment for untreated ($n = 27$), doxorubicin ($n = 25$), gemcitabine ($n = 28$), cisplatin ($n = 30$) and paclitaxel ($n = 31$) cohorts. **b** Fraction of cases showing progressive disease (PD), stable disease (SD), partial response (PR) and complete response (PR) across cohorts in **a** according to mRECIST criteria. **c** Kaplan–Meier of tumor volume doubling-free survival for PDX cohorts in **a**. Referenced to untreated, **Log-rank $p < 0.005$, ****Log-rank $p < 0.0001$. **d** Heatmap of BestAvgResponse from PDX cohorts in **a** for models that were evaluated with multiple agents ($n = 30$ PDX models, 1 untreated condition, 4 chemotherapies). Number of drugs patient received prior to xenotransplantation (above). Overall chemosensitivity classification called chemosensitive ($\geq 2$ CR/PR) or resistant ($\leq 1$ CR/PR) (below). **e** Correlation plot of number of drugs patient received prior to xenotransplantation and the average of BestAvgResponse calls in the PDX for all agents evaluated ($n = 30$). Values colored by overall chemosensitivity classification. **f** Kaplan–Meier analysis of progression-free survival (PFS) for patients whose PDXs were classified as overall chemosensitive ($n = 12$) versus chemoresistant ($n = 18$). **g** Waterfall plots for BestAvgResponse of PDXs treated with doxorubicin, cisplatin or paclitaxel, where responses are stratified based on patient response to that agent prior to xenotransplantation. *$p < 0.05$, ***$p < 0.001$. **h** Heatmap of patient and PDX response calls (sensitive CR/PR versus resistant SD/PD) based on patient response prior to xenotransplantation.

E545K in GCRC1971X) and *ERBB2* hotspot mutation (L755S in GCRC1715X) (Fig. 6a, Supplementary Table 2)[30,31]. Other potentially actionable alterations were *FGFR1*, *MYC* and *MDM2* amplifications, *ATM*, *ARID1A*, *CDH1* and *PIK3R1* truncating mutations and *PTEN* and *INPP4B* losses, which have demonstrated fair preclinical data[30,31]. Outlier expression analysis of RNA and RPPA data was also evaluated to identify further targets (Fig. 6a). Although this confirmed several of the clinically established targets (ER/PR and HER2/pHER2), it also revealed other expression outliers currently undergoing clinical (CD274/PD-L1, androgen receptor) and preclinical (pChk1/2/ATM/ATR, FASN/ACC1, FAK) evaluation[32,33].

Proof-of-concept functional experiments were performed for several models representing unique clinical challenges. For the GCRC1971X model, an ER + ILC that displayed skull-base metastases and multi-chemoresistance in vivo, WGS revealed multiple candidates (*PIK3CA* hotspot, *FGFR1* amplification, *CDH1* truncating mutation) (Figs. 4l–p and 6a). In addition to being amplified, *FGFR1* was highly expressed and was further pursued because of its known roles in ILC biology and endocrine therapy resistance, which the patient displayed (Fig. 6b)[34,35]. A small PCT was initiated to evaluate the efficacy of BGJ398, an orally available FGFR inhibitor, among a subset of PDXs displaying the highest *FGFR1* copy number and/or RNA expression across the PDX library (Fig. 6c). Although GCRC1971X achieved a CR within a week of initiating treatment, responses were poor for the four other models with lower levels of *FGFR1* amplification/expression (Fig. 6c, d). These findings were upheld in the metastatic setting, where BGJ398 induced regressions in spontaneous skull-base metastases for this PDX model (Fig. 6e).

To address the challenge of treating patients with rare histological variants, GCRC1979X was further investigated. This triple-negative neuroendocrine breast tumor was derived from a locally recurrent lesion one year after undergoing neoadjuvant chemotherapy and breast conserving surgery, at which point a PDX was developed. Consistent with the patient's lack of response to anthracycline/taxane, the PDX did not respond to doxorubicin or paclitaxel in the chemosensitivity screen (Fig. 5d). AKT was found to be an RPPA outlier, and given the success of everolimus in advanced gastrointestinal neuroendocrine tumors, which display frequent PI3K/mTOR activation, this pathway was

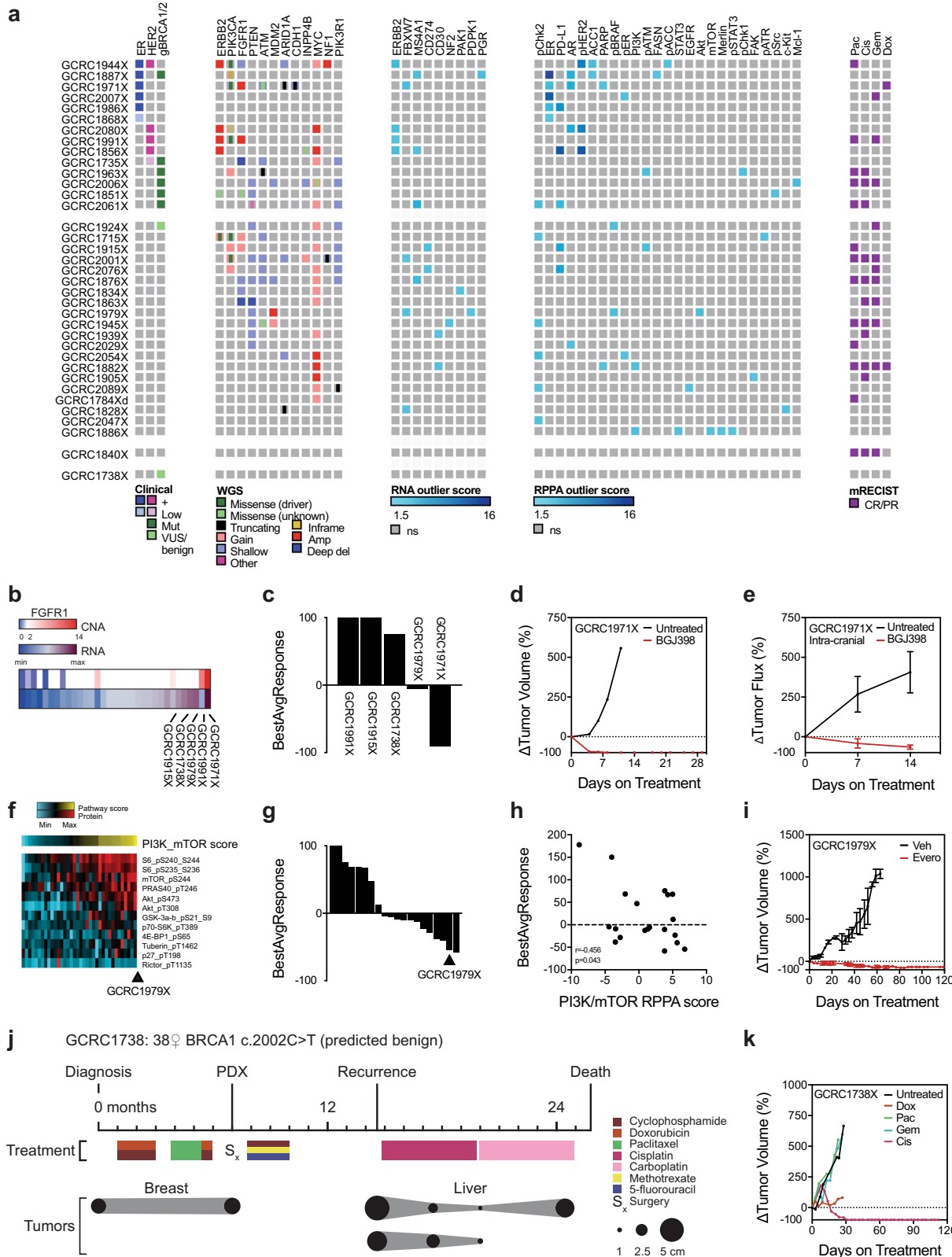

further interrogated[36]. The GCRC1979X PDX ranked highest for an RPPA-based PI3K/mTOR activation signature (Fig. 6f). The efficacy of everolimus was evaluated in the GCRC1979X model in the context of a PCT of 19 PDX models to further evaluate the specificity of response (Fig. 6g). Overall, everolimus response correlated with PI3K/mTOR activation score, where the

GCRC1979X displayed a strong and durable response to this agent (Fig. 6h, i).

Finally, GCRC1738X was used to illustrate the value of empirical in vivo drug testing in the adjuvant setting in a model where no clear actionable alterations were identified. This 38-year-old patient with TNBC and a germline *BRCA1* mutation

**Fig. 6 Chemogenomic profiling of PDXs reveals actionable feature for difficult-to-treat tumors. a** Heatmap of actionable features for PDX models ($n =$ 36) derived from clinical data (receptor and germline BRCA1/2 status), whole-genome sequencing (WGS; copy number alterations and mutations), outlier expression from RNA-sequencing and reverse-phase protein array (RPPA) and chemosensitivity from in vivo profiling. **b** Heatmap of *FGFR1* copy number and RNA expression across PDX library ($n = 33$). **c** Waterfall plot of BestAvgResponse for PDX clinical trial in mice treated with BGJ398 ($n = 5$). **d** Tumor growth curve for mammary fat pad tumors of GCRC1971 PDX either untreated or BGJ398-treated ($n = 1$ per arm). **e** Tumor growth curve for spontaneous luciferase-tagged skull-base metastases of GCRC1971 PDX untreated and BGJ398-treated mice ($n = 3$ per arm). Mean ± SEM. **f** Heatmap of PI3K/mTOR combined RPPA signature score (above) and expression of individual probes (below) across PDXs ($n = 36$). Samples ranked on PI3K/mTOR score. **g** Waterfall plot of BestAvgResponse for PDX clinical trial in mice treated with everolimus ($n = 20$). **h** Correlation plot of PI3K/mTOR RPPA score and BestAvgResponse to everolimus for cohort in **g**. **i** Tumor growth curve for GCRC1979 PDX either vehicle or everolimus-treated ($n = 3$ per arm). Mean ± SEM. **j** Schematic of clinical history for patient GCRC1738, including treatments and tumor sizes. **k** Tumor growth curves for GCRC1738 PDX either untreated, doxorubicin, paclitaxel, gemcitabine or cisplatin treated ($n = 1$ per arm).

(c.2002C>T) reported as benign, failed to respond to neoadjuvant doxorubicin, cyclophosphamide and paclitaxel (Fig. 6j). A PDX was generated at the time of surgery and chemosensitivty was evaluated as the patient underwent adjuvant chemotherapy (cyclophosphamide, methotrexate and 5-fluorouracil). Similar to the patient, the PDX did not regress with paclitaxel or doxorubicin (Fig. 6k). Seven months after surgery, the patient recurred with multiple liver metastases. Concordant with the response observed in the PDX, the patient was empirically started on cisplatin and demonstrated a marked regression of her large liver lesions (Fig. 6j, k). Eventually, the patient developed toxicity and required switching to carboplatin, and thereafter progressed. These functional studies demonstrate the potential for biomarker discovery and identification of precision therapies for diverse clinical challenges using a library of chemogenomically profiled PDXs.

## Discussion

Although tremendous progress has been made in the treatment of breast cancer leading to improved outcomes, there are subsets of patients who unfortunately remain particularly difficult-to-treat. Models that faithfully recapitulate both molecular and phenotypic features of these tumors are critical for the translation of basic discoveries to improve the management of these patients. The comprehensive molecular profiling and screening of metastatic potential and chemosensitivity herein indicate that PDXs largely maintain the biology of the tumors from which they are derived, and therefore represent a valuable preclinical resource.

The PDX library described herein fills several voids in the breast cancer modeling space. We have generated novel xenografts for tumor types where no models currently exist to our knowledge, including rare histological variants of breast cancer (e.g. NE, ACC). In addition, we describe the first breast PDX exhibiting highly selective skull-base organotropism (GCRC1971X). We have also contributed to a growing list of TNBC PDXs available to the research community, which will need to continue to expand to capture the full breadth of this heterogeneous subtype[37–40].

This collection also represents the largest series of distinct breast cancer PDX models, which have undergone WGS with their corresponding patient tumors. Although the mutational load, copy number profiles and driver alterations were robustly conserved, the strength of VAF correlations was variable across PDX lines (Fig. 2, Supplementary Fig. 5). Eirew et al.[15] have previously examined this using deep and single-cell sequencing to delineate multiple patterns of clonal dynamics upon xeno-transplantation. Given that we and others have demonstrated preservation of the transcriptional and protein signaling outputs in PDXs, the functional significance of these subclonal shifts remain unclear[12–14,41]. Furthermore, we find PDXs retain sig-naling activity, which have traditionally been associated with microenvironmental stimuli, including interferon and hypoxia

responses. Whether this is secondary to tumor–stromal interactions with the immunocompromised host mouse or cancer cell-autonomous properties remain unclear.

PDXs have recently been shown to serve as a robust platform for population-level preclinical drug screening using a high-throughput 1×1×1 approach[6]. Our data support the feasibility and reproducibility of this strategy, whereby PDX response rates to standard-of-care chemotherapeutics were similar to what is observed clinically (Fig. 5). Although measurement accuracy is sacrificed in search for clinically meaningful tumor regressions using hundreds of unique models in the 1×1×1 approach, we have also found value in evaluating drug sensitivity across ~30 PDX models with substantial molecular heterogeneity for signal detection in discovery-phase experiments. By screening a portion of this PDX cohort for gefitinib sensitivity, we have identified a subset of TNBCs, which highly express EGFR specifically within the tumorigenic subpopulation of their tumors, which renders them vulnerable to EGFR inhibition, as opposed to uniformly high EGFR expression, which did not correlate with sensitivity[42].

Defining the chemosensitivity landscape of our PDX library provides important clinical context in the evaluation of novel agents and may guide design for subsequent clinical trials. For example, drugs that cause regression in multi-chemoresistant PDX models may have enhanced success in the late-stage/meta-static setting versus another agent, which only shows activity in untreated/chemosensitive models, where the neoadjuvant setting may be more appropriate for clinical evaluation. Chemosensi-tivity profiles in conjunction with deep molecular characteriza-tion provides a platform for biomarker discovery and mechanistic studies. PDXs also allow for dynamic temporal sampling that can be a challenge in patients, a strategy that has been used to identify resistance mechanisms to PI3K inhibition in TNBC[43].

Our data are consistent with retention of chemosensitivity upon xenotransplantation, yet the prospect of using PDXs as avatars in a predictive setting to guide therapy will require co-clinical studies (e.g. REFLECT study, NCT02732860). Although cases representing clinical challenges were purposefully selected for inclusion in our study, chemogenomic profiling revealed nearly all models displayed actionable features that could be interrogated, several of which demonstrated marked regressions in vivo (Fig. 6). Although the drugs used in our study have been approved in other indications and/or safely used in humans, evaluating experimental agents in PDXs must account for species differences in pharmacokinetics/dynamics so that they are tested using clinically relevant dosing. Our data highlight that the practical application of avatars would be a challenge in the metastatic setting where patients often progress prior to estab-lishment of the PDX, however, the adjuvant setting in high-risk TNBCs could provide a window-of-opportunity where molecular characterization and functional testing could be performed prior to recurrence. Izumchenko et al.[11] have demonstrated that PDXs from multiple tumor types accurately recapitulate the patient's

clinical response even after late relapse and/or intervening lines of therapy, though further studies prospectively demonstrating the predictive capacity of PDXs are warranted. Although the broad applicability of this approach would currently be prohibited by cost and inefficient take rates, personalized avatars could be evaluated in the context of high-risk/difficult-to-treat patients as validation models for select drug candidates identified by higher-throughput explant cultures[16].

## Methods

**Tissue samples and patient-derived xenografts**. All tissue was collected with informed consent under REB-approved protocols at the McGill University Health Centre and Jewish General Hospital. Breast cancer patients over the age of 18 with (1) ER−; (2) HER2+; (3) high-grade ER+; (4) metastatic; or (5) rare histological variants of breast cancer undergoing diagnosis and/or management at McGill University Health Centre or Jewish General Hospital, Montreal, QC, Canada were recruited for this study. Mice were maintained and treated in accordance with the Facility Animal Care Committee at the Goodman Cancer Research Centre of McGill University (2014-7514). Excess breast tumor tissue was transported to the laboratory in ice-cold transport medium (DMEM/F12, 50 μg/ml gentamicin, 1× penicillin– streptomycin, 2.5 μg/ml fungizone). Samples were cut into 1 mm³ fragments and transplanted into the fourth mammary fat pad of 5–7-week-old NOD.Cg-*Prkdc*scid*Il2rg*tm1Sug/JicTac (NOG) female mice (Taconic) under sterile conditions. Fine needle aspirates were washed in PBS, resuspended in PBS:matrigel (Corning) (1:1) and 50 μl was injected orthotopically. For ER+ tumors, estrogen supplementation was administered as subcutaneous wax pellets as previously described[44]. Mice were palpated weekly and measured using calipers and harvested for in vivo passaging when tumors reached endpoint (>10 mm in the largest dimension). Tumors fragments were cryopreserved by freezing in FBS with 10% DMSO in a CoolCell container (Biocision) at −80 °C.

**Immunohistochemistry**. Tissues were fixed in 10% neutral-buffered formalin for 24 h, paraffin-embedded and sections were cut at 4 μm. Sections were deparaffinised in xylenes, re-hydrated in ethanol followed by antigen retrieval in boiling 10 mM citrate buffer (pH 6.0). Slides were blocked with Power Block (BioGenex), incubated primary antibody for 1 h at room temperature with the following antibodies: ER (SP1, 790-4324), HER2 (4B5, 790-2991), Ki67 (30-9, 790-4286), Pan-Keratin (AE1/AE3/PCK26, 760-2135), Vimentin (V9, 790-2917), p53 (DO-7, 790-2912), CK5/6 (D5/16B4, 790-4554), CK8/18 (B22.1 & B23.1, 760-4344), CD45 (RP2/18, 760-2505), p63 (4A4, 790-4509), Synaptophysin (SP11, 790-4407) or E-cadherin (36, 790-4497) (Ventana). This was followed by 3% H₂O₂ for 30 min then by SignalStain Boost (Cell Signaling) secondary antibody for 30 min. The SignalStain DAB substrate kit (Cell Signaling) was used as a detection method prior to counterstaining with Harris' hematoxylin, dehydration and mounting. Slides were scanned using Aperio-XT slide scanner (Aperio). Sankey diagrams were constructed with SankeyMATIC (BETA).

**DNA and RNA isolation**. Tissue was snap frozen in OCT (Tissue-Tek) and sectioned for histopathological review. Total DNA and RNA was isolated from adjacent sections using the AllPrep DNA/RNA Mini Kit (Qiagen). Germline DNA was derived from buffy coat and extracted using the DNA Blood Maxi Kit (Qiagen). DNA was quantified using the Qubit fluormeter. RNA was quantified by NanoDrop and integrity was evaluated with Bioanalyzer 2100 (Agilent). The following samples were not included in genomic analyses: GCRC1924T (WGS/RNA-seq; tumor sample unavailable), GCRC1944T (RNA-seq; low-quality RNA), GCRC1979T (WGS/RNA-seq; tumor sample unavailable), GCRC1986T (WGS/RNA-seq; tumor and germline samples unavailable), GCRC1986X (WGS; patient germline sample unavailable for alignment), GCRC2054T (WGS/RNA-seq; tumor sample unavailable), GCRC2076T (WGS/RNA-seq; tumor sample unavailable), GCRC2089T (RNA-seq; low-quality RNA).

**Whole-genome and RNA-sequencing**. WGS libraries were prepared using PCR-free construction and sequenced at 1 lane per sample on HiSeq X with v3 chemistry according to Illumina protocols, generating 150 bp paired-end reads. RNA-seq libraries were prepared using ssRNA-seq construction and sequenced on HiSeq2000 according to Illumina protocols, generating 75 bp paired-end reads.

**WGS analysis**. Short paired-end reads were trimmed using Trimmomatic (version 0.35)[45] and the resulting reads were aligned to the GRCh38/hg38 human reference genome using BWA-MEM (version 0.7.15)[46]. Alignments were recalibrated by using GATK (version 3.8)[47] and duplicates were marked with Picard (version 2.9; http://broadinstitute.github.io/picard/). To remove possible contaminated reads originating from mouse in xenograft samples, reads were also aligned to the GRCm38/mm10 mouse and the Disambiguate algorithm (version 1.0)[48] was used to assign the reads to individual species based on the highest quality alignment of the read pair. Somatic mutations were identified by GATK's Mutect2 algorithm[49] at default parameters in each tumor and xenograft sample using its matched

normal tissue as reference. Structural variants were called using Manta at default parameters (version 1.5)[50]. Copy number variation analysis was performed using SCoNEs (version 2.1; https://bitbucket.org/mugqic/scones/) with a bin size parameter of 10 kb. Variants were annotated with hg38 database using SnpEff (version 4.3)[51]. CRAVAT 5.2.4 was used to identify potential pathogenic and cancer driver variants[52]. Genes previously associated in breast cancer were retrieved from Nik-Zainal et al.[19]. Oncoprints were generated from cBioPortal (https://www.cbioportal.org/).

**RNA-seq analysis**. Adaptor sequences and low-quality score bases (Phred score < 30) were first trimmed using Trimmomatic[45]. The resulting reads were aligned to the human genome reference sequence (GRCh38/hg38), using STAR[53]. Reads originating from mouse in xenograft samples were removed using the Disambiguate algorithm[48]. Read counts for each sample are obtained using HTSeq[54]. For downstream analyses, lowly-expressed genes with an average read count lower than 10 across all of the samples were excluded. Raw counts were normalized using the TMM algorithm (i.e., weighted trimmed mean of *M*-values), implemented in edgeR R package (version 3.22.5)[55]. Using the voom function in the limma R package (version 3.36.5)[56], the data was converted to log-counts per million with associated precision weights. The lmfit function from limma was used to identify differences in expression levels between primary tumor and xenograft models with the following paired design: Expression ~ Individual + Model. Nominal *p*-values were corrected for multiple testing using the Benjamini–Hochberg method. Pathway analysis was performed with single-sample gene set enrichment analysis (v4) using GenePattern3.9.10[57,58]. Heatmaps were constructed using Morpheus (https://software.broadinstitute.org/morpheus).

**Reverse-phase protein array**. Snap frozen tissue was analyzed by RPPA with 244 antibodies as previously described[33]. Briefly, lysates were serially diluted, arrayed onto nitrocellulose-coated slides, and probed with antibodies using a tyramide-based signal amplification approach followed by a DAB colorimetric reaction. Slides were scanned, spots were quantified by Array-Pro Analyzer and relative protein levels were determined by interpolation of each dilution curve from the standard curve (SuperCurve Rx64 3.1.1) of the slide. Data was normalized for loading and transformed to linear values or median-centered log2 values. Signature scores were generated by summing positive regulatory components minus negative regulatory components on median-centered log2 values from previously published RPPA signatures[33]. GCRC1784Xc (derived from a metaplastic tumor with chondroid differentiation) was not included in the analysis because it was an outlier (Correction Factor 1: 0.33) sample for total protein content. The TCGA Breast Invasive Carcinoma, Firehose Legacy RPPA and RNA-seq datasets were downloaded from cBioPortal (https://www.cbioportal.org/) on January 27, 2020.

**Actionability analysis**. Actionable genomic alterations were retrieved from OncoKB (actionable genes restricted to all, solid or breast tumors; downloaded on May 22, 2019) and ESMO/ESCAT[30,31]. Actionable RNA and RPPA expression outliers were retrieved from DEPO (http://dinglab.wustl.edu/depo; accessed on May 22, 2019) and DEPO/Akbani et al.[32,33], respectively. Actionable alterations are listed in Supplementary Table 2. Outlier analysis was performed as previously described[41]. Briefly, an outlier score was calculated for each gene/protein as (x-Q3)/IQR (Q3, third quartile; IQR, interquartile range), and called an outlier if the value was >1.5.

**In vivo drug sensitivity**. In vivo drug sensitivity studies were done in 5–7-week-old NOD.Cg-*Prkdc*scid *Il2rg*tm1Wjl/SzJ (NSG) female mice (Jax) using a 1×1×1 approach[6]. Briefly, tumor fragments orthotopically engrafted in mice were allowed to grow to 100 mm³ before initiating a 28-day treatment regimen. The following drug regimens were used: 3 mg/kg doxorubicin (in 0.9% normal saline) intravenous (IV) weekly, 24 mg/kg paclitaxel (in 1:1:18 Kolliphor:Ethanol:0.9% normal saline) IV weekly, 4 mg/kg cisplatin (in 0.9% normal saline) IV weekly, 50 mg/kg gemcitabine (in 0.9% normal saline) IV twice per week, 7.5 mg/kg everolimus (in 10% NMP, 90% PEG300) oral daily, 30 mg/kg BGJ398 (in 1:1 acetate buffer, pH 4.6: PEG300) oral daily. Tumor dimensions were measured twice per week, volume was calculated according to the formula $V = (length \times width^2)/2$. BestAvgResponse and response calls (CR, PR, SD, PD) were calculated as previously described[6]. Briefly, response was determined by comparing tumor volume change at time *t* to its baseline with the formula $\Delta V_t = ((V_t - V_{initial})/V_{initial}) \times 100$. The BestAvgResponse was calculated as the minimum of the average of $\Delta V_t$ from $t = 0$ to $t$ for $t \geq 14$ days.

**Spontaneous metastasis models**. The GCRC1971 PDX was luciferase-tagged by infecting short-term cultures. Briefly, PDX tumors were dissociated at 37 °C for 1 h on a rotisserie in digestion medium (RPMI, 2.5% FBS, 10 mM HEPES, 1 mg/ml collagenase type IV, 50 μg/ml gentamicin), with brief vortexing and trituration every 15 min. Single-cell suspensions were generated with 0.25% trypsin/EDTA (Gibco) for 5 min at 37 °C and dispase/DNase (STEMCELL), then passed through a 40 μm strainer. Cells were seeded in suspension in tumorsphere medium (DMEM/F12, 1× B27, 20 ng/ml human EGF, 10 μg/ml insulin, 0.5 mg/ml hydrocortisone, 20 ng/ml bFGF, 10 μg/ml heparin, 50 μg/ml gentamicin) on ultra-low attachment

plates (Corning). The following day, lentiviral pHIV-Luc-ZsGreen (Addgene) particles were added for 8 h with 1× polybrene. Two days following infection, cells were thoroughly washed and $1 \times 10^5$ to $1 \times 10^6$ cells in PBS:Matrigel (1:1) were injected into the mammary fat pad of NOG mice. Tumors were resected at 10 mm, and mice were monitored using luciferin and the IVIS Spectrum system.

**Statistics and reproducibility.** Measurements were taken from distinct samples when applicable. No sample size calculations were performed. Prism 8 (GraphPad) was used for basic statistical analysis. Significance was determined using Student's t-test (two-tailed), Fisher's exact test, Pearson correlation and survival analysis using the Log-rank (Mantel–Cox) test. Benjamini–Hochberg correction was applied where noted in the text. p-value < 0.05 was considered significant.

**Reporting summary.** Further information on research design is available in the Nature Research Reporting Summary linked to this article.

## Data availability

Whole-genome sequencing data has been deposited in the Sequence Read Archive under the accession PRJNA594000. RNA-sequencing data has been deposited in the Gene Expression Omnibus under the accession GSE142767. Any remaining data pertaining to this manuscript is available from the corresponding author upon reasonable request.

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

## Acknowledgements

We thank the patients that provided tissue for this study; the MUHC breast surgeons and pathologists for assisting with clinical specimen procurement; the Goodman Cancer Research Centre Histology Facility for histological processing; the Michael Smith Genome Sciences Center for sequencing; and the MD Anderson RPPA core facility. P.S. and M.D. are Vanier Scholars and the recipients of CIHR MD/PhD studentships. M.P. holds the Diane and Sal Guerrera Chair in Cancer Genetics and P.M.S. is a McGill University William Dawson Scholar. This work was supported by the Réseau de Recherche en Cancer of the FRQS, Québec Breast Cancer Foundation, CIHR Foundation Grant and SU2C (to M.P.).

## Author contributions

P.S. and M.P. conceived the study; P.S., V.P., A.A-.M. and M.G.A. generated patient-derived xenografts; P.S., L.L., V.P., A.A-.M., M.D. and C.M. performed mouse experiments; A.W., H.Z. and M.S. extracted and prepared nucleic acids; P.S., A.P., H.K. and D.L. performed bioinformatic analyses; V.M-.R., A.A-.M., J.L. and N.R.B. assisted with patient recruitment; J.A., N.B., K.P., A.O., M.B. and S.M. provided access to patient material; P.S., P.M.S., S.P.S., S.A., M.B., S.M. and M.P. provided study supervision; P.S. and M.P. wrote the manuscript; all authors approved the manuscript.

## Competing interests

S.A. is the cofounder and advisor to Contextual Genomics and scientific advisor to Sangamo Therapeutics, Chordia Therapeutics and Repare Therapeutics. All other authors declare no competing interests.
