## [Peer Review File · Communications Biology]

Reviewers' comments:

Reviewer #1 (Remarks to the Author):

Savage et al. used 37 breast cancer patient-derived xenografts (PDX) models to complete whole-genome, transcriptome sequencing and RPPA to identify potential therapeutic targets, metastatic potential and chemosensitivity across all tumors. In addition to PDX characterization, the authors also go on to complete proof-of-concept validation experiments for several models to show the potential clinical application of this method for personalized medicine.

Major

- Please clarify the following sentence (line 185) "The sequence context of SNVs, reflecting distinct mutational processes, was well preserved in PDXs"

- Figure 3:

- o The red/green colormap is not colorblind friendly

- o Can you add in a subtype colorbar to 3H to see how these pathways vary by subtype?

- o 3B: Do these pathways group into distinct pathway clusters? Can some of the pathways of interest be labeled on B to highlight some of the more interesting findings? Overlay subtypes or other clinical features as a colorbar on top to see what drives the clustering? Otherwise this figure doesn't provide much interesting information.

- Figure 4:

- o Can you make it clearer that each of the bars in E, I, M is a different mouse?

- Paragraph starting on line 262: Many studies have looked at the correlation between RPPA data and RNA-Seq. Are the TCGA breast tumor results consistent with your results? Do you find the same probes correlate poorly with your RNA expression?

- The manuscript never references Mundt et al., Mass Spectrometry-Based Proteomics Reveals Potential Roles of Resistance to PI3K Inhibition in Triple-Negative Breast Cancers" Cancer Research 2018, which did a similar type of personalized medicine approach using PDXs with global proteomics as opposed to RPPA. I think it would be worth noting this study in the introduction or discussion and how it relates to their work.

Minor

- Figure 1D is quite difficult to interpret with all of the overlaid information

- Figure S5 – the chromosome numbers are unreadable, I would add a legend somewhere so people can evaluate the chromosomal rearrangements more easily

- On line 315: "... 28-day study across 25-31 PDX lines, respectively". The respectively here doesn't make sense since you're referring to 4 drugs and provide one range.

Reviewer #2 (Remarks to the Author):

The manuscript describes the characterization of a set of breast PDXs derived by the authors. Most of the PDXs are derived from patients having received multiple rounds of treatment and represent a more aggressive subset of breast cancer. A comprehensive body of work has been clearly presented and the PDX bank should provide a useful resource. However, the findings relate more to this bank than the wider field as there does not seem to be any exploration of novel mutations, pathway interactions, novel drug targets or testing of novel drugs.

Specific comments:

1.The proposition that PDXs could be use in prospective personalized drug sensitivity testing (lanes 137-138) is not really supported and the only prospective testing (Fig. 6J) using PDXs is carried out on those from a patient recurrence (which might have evolved from the primary lesion). Indeed, one of the hurdles in such testing is the length of time in the development and analysis of PDXs.

2.There is a discrepancy in the number of PDXs/patients noted throughout the manuscript without explanation. It would be worth noting why tissues have been omitted:

Lanes 118-119: 37 PDXs from 36 tumors from 29 patients

Lanes 134-136 (relating to panel 1D): 24 + 11 makes 35

Lane 179: 36 PDXs

Lane 192: 34 PDX-primary pairs

Lane 228: 37 PDXs and 29 matched patient tumors

Lane 253: 37 PDX

Lane 640: 36 PDX, 31 patient tumors

Lane 13 Supp fig: 28 models

3.Explanation of panel 1D is confusing when looking at the data. As pointed above 24 + 11 is 35 and there are 36 columns. There are 25 pre-treated black squares on the second row of the heatmap which should correspond to recurrent/metastatic disease but the number in the text is 7/11. Conversely, the 3/24 does not fit with the 11 white squares. This section needs to be reviewed as well as the conclusion at the end of the paragraph.

4.Missing scale bars in Fig.4B, S2, S4A.

5.There are at least 6 BRCA1 carrier patients (nearly 1/6) of the total, most of them treatment naïve. These are a valuable resource. Have the authors thought about looking at those as a separate subset?

Reviewer #3 (Remarks to the Author):

Review NatureComms 02/2020

Comments for author

The use of PDXs in translational research has gain much attention in the past years. However, the best use of such state-of-the-art models in drug development and clinical practise is still being evaluated. As such, albeit other well annotated breast cancer PDX cohorts have been previously published and support the current manuscript observations, here, the authors very elegantly and deeply molecularly and histologically characterised a cohort of breast cancer PDX of "aggressive" and "rare" subtypes of breast cancer, which are indeed those that require more attention.

As mentioned above, much of the data published here supports previous literature, and hence lacks novelty. The authors should however highlight their new very exciting and promising observations even further. For example, the use of PDXs as Avatars to inform on real time clinical decision making is intriguing indeed. This manuscript brings some hope to a possible use of PDXs in anticipating drug responses in a subset of breast cancer samples, of strong clinical implications, the TNBCs in the adjuvant setting (those examples that recur after PDX P2 have been generated... and tested?).

The figures are self-explanatory, layouts and illustrations elegantly presented and most showing the actual experimental data. I would suggest the raw data on the chemosensitivity preclinical trials to be shown in a more transparent way.

Caution should be made when interpreting the results on the experiments done to check the retention of clinical drug responses in PDXs. There are several confounding factors and the authors should make it even clearer when they are studying the "Retention" of a retrospective response rather and when the "prediction" of a future one in case patients' relapse. The authors should also comment on the discussion that the best way would be to study this in parallel in the form of a co-clinical trial for example.

More specific comments

Figure 1 and text

Needs to be accompanied with some mention to previous literature that supports and is in line with their findings.

Which is the novelty at the molecular characterization level? Or is it supporting literature? Is the novelty mainly on the generation of preclinical models from rare subtypes? If so, it is all very good but should all be cleared in the main text too.

The authors perform QC analysis on the current PDX biobank based on known spontaneous patient immune-cell derived grafts. Did the authors also checked for spontaneous mouse tumours , especially for those with really fast growth rates? Spontaneous mouse tumours can arise at any given point in the family line of a given model. Proof of concept established

Also, some genotyping to confirm a match to the human originating sample would be useful.

Is there an association between P1/P2 growth rates and time to progression?

In the discussion, this sentence "In other tumor types, the initial PDX maintained its predictive capacity even after late relapse and/or intervening lines of therapy (11)" would need to go linked to something like "needs further investigation"

Major comments

There is heterogeneity in drug responses amongst the PDXs tested, which seems to correlate with observations in the clinic. Although number model-drug tests are not high enough to make this conclusion, plus I don't think is the scope nor the relevance of this manuscript (and has been done before by Gao et al. Nat Med 2015). Overall, PDXs seem to be responding less to Dox than to Pacli. The manuscript indeed shows PDXs do not respond to Dox, regardless of the response to the patient. The authors do mention this could be due to insufficient drug concentration used for the Dox trials due to mouse-specific toxicity. If so, other confounding factors related to drug PK/PD could be leading to misleading interpretations, consistent with the still high discordant results between in vivo and clinical responses. I understand it is difficult to test the match drug response clinical data in patients. The authors should at least discuss this clearly in the discussion (including possible differences in metabolisms of the drug, differences in PK/PD affecting drug's blood and tumour concentrations, the difficulty of comparing to match single agent chemo treatment in patients, and the difference between "retaining" to "predicting" ...).

The manuscript main observation in my opinion is that their data suggests heavily treated tumours are generally less sensitive to further lines of chemo treatment, and their data on concordance and discordance is weaker, also because logic follows that treatment will change their drug response profile. Do they have any match PDX treatment naïve vs post-treatment or pre and post-treated that could be tested? And hence highlight the relevance of such effect? Or could this be treatment specific?

How do you interpret VAF changes? $R < 0.3$ indicates a strong clonal drift... how is this consistent with previous literature? And is there anything that explains such heterogeneity in VAF dynamics upon xenotransplantation in these models?

The "one mouse per treatment per model" approach is not the best approach to test drug responses in few models. However, I understand the financial and logistic implications of doing a 3x3 approach, plus the authors show when using the later, the results were very similar. A mention in the discussion that such 1x1x1 approach is better suited for large scale PDX preclinical trials which compensate the loss of measurement accuracy on individual mice would be good...

"This classification was consistent with clinical outcomes, where patients with chemoresistant PDXs displayed worse PFS... "... this is a circular argument. Chemo resistant PDXs are those from heavily treated patients which are likely to be at a more progressive state of the disease, and hence have a worse prognosis.

The fidelity of the PDX response was assessed as a clinical cohort and from the perspective of an individual patient. At the population level—ref 23-25.. has the rate of response to single agents been evaluated in a clinical cohort? And is the clinical cohort similar to the preclinical one? If not, this is an overstatement.

"The mechanism by which three cases went from paclitaxel resistant in the patient to sensitive in the PDX remains unclear, though the effect of drug holiday during PDX generation resulting in re-sensitization may play a role". This has been observed in other labs. It could also be because engraftment is done in treatment naïve mice, allowing the highly proliferating cells killed by chemo to regrow and repopulate the bulk of the tumour.

Reviewers' comments:

Reviewer #1 (Remarks to the Author):

Savage et al. used 37 breast cancer patient-derived xenografts (PDX) models to complete whole-genome, transcriptome sequencing and RPPA to identify potential therapeutic targets, metastatic potential and chemosensitivity across all tumors. In addition to PDX characterization, the authors also go on to complete proof-of-concept validation experiments for several models to show the potential clinical application of this method for personalized medicine.

Major

- Please clarify the following sentence (line 185) “The sequence context of SNVs, reflecting distinct mutational processes, was well preserved in PDXs”

Author response: *We have clarified this sentence and added a reference as follows: “The subtype of base substitutions (eg. C>A, C>G, C>T, T>A, T>C, T>G) were well-preserved in PDXs (Fig. 2A)(20).”*

- Figure 3:

- o The red/green colormap is not colorblind friendly

Author response: *The coloration of all the RNA-seq heatmaps (Fig. 3A, 3C-F, 6B, S7A, S7B) are now colorblind friendly (blue-red).*

- o Can you add in a subtype colorbar to 3H to see how these pathways vary by subtype?

Author response: *The subtype colorbar has been added to Fig. 3H (now Fig. 3F).*

- o 3B: Do these pathways group into distinct pathway clusters? Can some of the pathways of interest be labeled on B to highlight some of the more interesting findings? Overlay subtypes or other clinical features as a colorbar on top to see what drives the clustering? Otherwise this figure doesn't provide much interesting information.

Author response: *Colorbars for clinicopathologic parameters and specific pathways of interest have been highlighted in Fig. 3B. We have emphasized that some of these pathways are associated with subtype, while others are not in both the figure and in the text. We have modified the text as follows: “Pathway-level analysis was performed using single sample gene set enrichment analysis (ssGSEA) across 2,117 pathways (‘C2 chemical and genetic perturbations’ pathways from mSigDB)(Fig. 3B)(23). Unbiased clustering of pathway activation appeared to be largely driven by subtype, where luminal, ERBB2 and basal expression modules were well-preserved (Fig. 3B-C). In addition to this, other pathways (eg. cell cycle, interferon response, hypoxia) demonstrated variability within subtype. Despite distinct species microenvironments, significant correlations were observed for pathway activation scores between human tumor and PDXs for key*

pathways associated with breast tumorigenesis, including proliferation, hypoxia and EMT (Fig. 3B,D)."

- Figure 4:

- o Can you make it clearer that each of the bars in E, I, M is a different mouse?

Author response: *We have clarified the figure legend for Fig. 4E,I,M as follows: "E. Bar chart of time of resection (at 10 mm) and endpoint for each PDX animal (n=9). Bars represent individual mice."*

- Paragraph starting on line 262: Many studies have looked at the correlation between RPPA data and RNA-Seq. Are the TCGA breast tumor results consistent with your results? Do you find the same probes correlate poorly with your RNA expression?

Author response: *The RPPA vs. RNA-seq R values were similar in our dataset with what was observed in the TCGA. We have added this analysis to Supplementary Fig. 7E and the following to the text: "This was consistent with TCGA data, where a significant correlation between RPPA/RNA-seq R values was observed in the 108 probes common to our and TCGA datasets (Supplementary Fig. 7E)."*

- The manuscript never references Mundt et al., Mass Spectrometry-Based Proteomics Reveals Potential Roles of Resistance to PI3K Inhibition in Triple-Negative Breast Cancers" Cancer Research 2018, which did a similar type of personalized medicine approach using PDXs with global proteomics as opposed to RPPA. I think it would be worth noting this study in the introduction or discussion and how it relates to their work.

Author response: *We have added this reference into the discussion in the following context: "PDXs also allow for dynamic temporal sampling that can be a challenge in patients, a strategy that has been used to identify resistance mechanisms to PI3K inhibition in TNBC (43)."*

Minor

- Figure 1D is quite difficult to interpret with all of the overlaid information

Author response: *We have simplified Fig. 1D by splitting the information into distinct panels. Fig. 1D now only focuses on the growth kinetics, while Fig. 1F are Sankey diagrams highlighting concordance rates of ER, HER2, histology and subtype status in patient and corresponding PDXs.*

- Figure S5 – the chromosome numbers are unreadable, I would add a legend somewhere so people can evaluate the chromosomal rearrangements more easily

Author response: *We have added chromosome colors to the legend.*

- On line 315: "... 28-day study across 25-31 PDX lines, respectively". The respectively here doesn't make sense since you're referring to 4 drugs and provide one range.

Author response: *“Respectively” has been removed from this sentence.*

Reviewer #2 (Remarks to the Author):

The manuscript describes the characterization of a set of breast PDXs derived by the authors. Most of the PDXs are derived from patients having received multiple rounds of treatment and represent a more aggressive subset of breast cancer. A comprehensive body of work has been clearly presented and the PDX bank should provide a useful resource. However, the findings relate more to this bank than the wider field as there does not seem to be any exploration of novel mutations, pathway interactions, novel drug targets or testing of novel drugs.

Specific comments:

1. The proposition that PDXs could be used in prospective personalized drug sensitivity testing (lanes 137-138) is not really supported and the only prospective testing (Fig. 6J) using PDXs is carried out on those from a patient recurrence (which might have evolved from the primary lesion). Indeed, one of the hurdles in such testing is the length of time in the development and analysis of PDXs.

Author response: *We agree that growth kinetics of breast PDXs may be a barrier to their implementation as avatars, particularly in the setting of advanced disease (Fig. 1D). We have emphasized this by re-wording this sentence: “This highlights a potential barrier for prospective personalized drug sensitivity testing using PDXs as avatars in advanced breast cancer.”*

2. There is a discrepancy in the number of PDXs/patients noted throughout the manuscript without explanation. It would be worth noting why tissues have been omitted:

Author response: *We have addressed the specific comments below and have also clarified these missing samples in the Methods section under DNA and RNA isolation: “The following samples were not included in genomic analyses: GCRC1924T (WGS/RNA-seq; tumor sample unavailable), GCRC1944T (RNA-seq; low quality RNA), GCRC1979T (WGS/RNA-seq; tumor sample unavailable), GCRC1986T (WGS/RNA-seq; tumor and germline samples unavailable), GCRC1986X (WGS; patient germline sample unavailable for alignment), GCRC2054T (WGS/RNA-seq; tumor sample unavailable), GCRC2076T (WGS/RNA-seq; tumor sample unavailable), GCRC2089T (RNA-seq; low quality RNA).”*

Lanes 118-119: 37 PDXs from 36 tumors from 29 patients

Author response: *This line actually reads “37 novel PDX lines derived from 36 tumors from 34 unique patients.” This is explained in the following two sentences in the manuscript: “One patient had three PDX models derived from their tumors – two sublines from distinct histological regions from their primary tumor (GCRC1784Xd/c, discussed below) and one from mediastinal lymph node metastasis (GCRC2054X) that developed at a later time point. Another patient had two PDXs developed from their tumors – one*

from their primary tumor (GCRC1915X) and another from a lung metastasis (GCRC2076X), which was sampled at the time of recurrence.”

This is also presented in Table 1 in the manuscript and re-iterated in the following table:

Patient	Tumor sample	PDX model
GCRC1784/2054	GCRC1784T (primary)	GCRC1784Xc (chondroid histological component)
	GCRC1784T (primary)	GCRC1784Xd (ductal histological component)
	GCRC2054 (mediastinal lymph node metastasis)	GCRC2054X
GCRC1915/2076	GCRC1915T (primary)	GCRC1915X
	GCRC2076T (lung metastasis)	GCRC2076X

Lanes 134-136 (relating to panel 1D): 24 + 11 makes 35

Author response: 2/37 PDX models were not included in the growth kinetic for the following reasons:

-GCRC1784X: GCRC1784T was implanted into P1 mice and multiple histological sublimes developed (see Fig. S4B,C). We only included the GCRC1784Xd subline because it had P1 mice with pure ductal histology, in contrast with GCRC1784Xc which only had a P1 mouse with mixed chondroid and ductal histology and therefore was excluded from this analysis.

-GCRC2089X: there were not sufficient P2 mice ($n < 3$) for growth kinetic data.

Lane 179: 36 PDXs

Author response: 1/37 PDX models was not included in the WGS analysis for the following reason:

-GCRC1986X: patient germline sample unavailable for sequence alignment.

Lane 192: 34 PDX-primary pairs

Author response: 3/37 germline-primary-PDX trios were not included in the WGS analysis for the following reasons:

-GCRC1986T: patient tumor and germline samples unavailable

-GCRC1924T: patient tumor sample unavailable

-GCRC1979T: patient tumor sample unavailable

Lane 228: 37 PDXs and 29 matched patient tumors

Author response: All PDXs were included and 7/36 patient samples were not included in the RNA-seq analysis for the following reasons:

-GCRC1986T: patient tumor sample unavailable
-GCRC1924T: patient tumor sample unavailable
-GCRC1944T: patient tumor sample had low quality RNA
-GCRC1979T: patient tumor sample unavailable
-GCRC2054T: patient tumor sample unavailable
-GCRC2076T: patient tumor sample unavailable
-GCRC2089T: patient tumor sample had low quality RNA

Lane 253: 37 PDX

Author response: All PDX samples were subject to RPPA, however, 1/37 PDX models were not included in RPPA analysis for the following reason:

-GCRC1784Xc: outlier sample (Correction Factor 1 = 0.33) for total protein content

Lane 640: 36 PDX, 31 patient tumors

Author response: 1/37 PDX and 5/36 patient tumors were not included in WGS analysis:

-GCRC1986X: patient germline sample unavailable for sequence alignment
-GCRC1986T: patient tumor and germline samples unavailable
-GCRC1924T: patient tumor sample unavailable
-GCRC1979T: patient tumor sample unavailable
-GCRC2054T: patient tumor sample unavailable
-GCRC2076T: patient tumor sample unavailable

Lane 13 Supp fig: 28 models

Author response: 9/37 PDX models did not have complete P1-3 growth kinetic data and were therefore not included in this analysis, leaving 28 PDX models. This was largely due to insufficient P3 mice to analyze P3 growth kinetics, as seen in Fig. 1D.

3. Explanation of panel 1D is confusing when looking at the data. As pointed above 24 + 11 is 35 and there are 36 columns. There are 25 pre-treated black squares on the second row of the heatmap which should correspond to recurrent/metastatic disease but the number in the text is 7/11. Conversely, the 3/24 does not fit with the 11 white squares. This section needs to be reviewed as well as the conclusion at the end of the paragraph.

Author response: Based on another reviewer's comments, we have removed the heatmap component of Fig. 1D as it was confusing. This information is still in Table 1 and is also illustrated in Fig. 1F as Sankey diagrams. Still, the inconsistencies are addressed below:

24+11=35 vs 36 columns:

GCRC1784Xd was not included for the reasons described above.

GCRC2089X is depicted in Fig 1D but not in the analysis in the text because there were insufficient mice ($n < 3$) for P2 data.

We should also note that pre-treatment and recurrence/metastasis are not synonymous because many of these models were generated at the time of surgery after neoadjuvant chemotherapy.

4. Missing scale bars in Fig.4B, S2, S4A.

Author response: *We have added scale bars to Fig. 4B, S2 and S4A.*

5. There are at least 6 BRCA1 carrier patients (nearly 1/6) of the total, most of them treatment naïve. These are a valuable resource. Have the authors thought about looking at those as a separate subset?

Author response: *We agree that these BRCA1 mutated models represent a valuable resource and are likely enriched in this study given our selection criteria. We have ongoing collaborative projects evaluating novel agents specifically for this subset of patients.*

Reviewer #3 (Remarks to the Author):

Review NatureComms 02/2020

Note: *We have provided point-by-point answers to these individual comments below, but collectively have decided to re-format the Discussion in our manuscript to address this reviewers constructive comments.*

Comments for author

The use of PDXs in translational research has gain much attention in the past years. However, the best use of such state-of-the-art models in drug development and clinical practise is still being evaluated. As such, albeit other well annotated breast cancer PDX cohorts have been previously published and support the current manuscript observations, here, the authors very elegantly and deeply molecularly and histologically characterised a cohort of breast cancer PDX of “aggressive” and “rare” subtypes of breast cancer, which are indeed those that require more attention.

As mentioned above, much of the data published here supports previous literature, and hence lacks novelty. The authors should however highlight their new very exciting and promising observations even further. For example, the use of PDXs as Avatars to inform on real time clinical decision making is intriguing indeed. This manuscript brings some hope to a possible use of PDXs in anticipating drug responses in a subset of breast cancer samples, of strong clinical implications, the TNBCs in the adjuvant setting (those examples that recur after PDX P2 have been generated... and tested?).

The figures are self-explanatory, layouts and illustrations elegantly presented and most showing the actual experimental data. I would suggest the raw data on the chemosensitivity preclinical trials to be shown in a more transparent way.

Author response: *We have now added the chemosensitivity tumor growth curves to Supplementary Fig. 9C and BestAvgResponse values are also now available in Supplementary Table 1.*

Caution should be made when interpreting the results on the experiments done to check the retention of clinical drug responses in PDXs. There are several confounding factors and the authors should make it even clearer when they are studying the “Retention” of a retrospective response rather and when the “prediction” of a future one in case patients’ relapse. The authors should also comment on the discussion that the best way would be to study this in parallel in the form of a co-clinical trial for example.

Author response: *We have emphasized that this analysis represents retention of drug response as opposed to prediction. We have changed the subtitle of this section to “PDXs retain the chemosensitivity profile of patients”. We have also referenced ongoing co-clinical trials using PDX avatars in the discussion: “Our data is consistent with retention of chemosensitivity upon xenotransplantation, yet the prospect of using PDXs as avatars*

in a predictive setting to guide therapy will require co-clinical studies (eg. REFLECT study, NCT02732860)."

More specific comments

Figure 1 and text

Needs to be accompanied with some mention to previous literature that supports and is in line with their findings.

Author response: *We have now stated that the data from Figure 1 is consistent with previous breast PDX cohorts: "Together, the engraftment success rates, association with clinicopathologic factors and outcomes as well as growth kinetics described herein are consistent with previously published breast PDX cohorts (12,13,15)."*

Which is the novelty at the molecular characterization level? Or is it supporting literature? Is the novelty mainly on the generation of preclinical models from rare subtypes? If so, it is all very good but should all be cleared in the main text too.

Author response: *We have highlighted some of the points of novelty and supportive aspects of our work in the discussion as follows: "The PDX library described herein fills several voids in the breast cancer modeling space. We have generated novel xenografts for tumor types where no models currently exist, including rare histological variants of breast cancer (eg. NE, ACC). Additionally, we describe the first breast PDX exhibiting highly selective skull-base organotropism (GCRC1971X). We have also contributed to a growing list of TNBC PDXs available to the research community, which will need to continue to expand to capture the full breadth of this heterogeneous subtype (37–40).*

This collection also represents the largest series of distinct breast cancer PDX models which have undergone WGS with their corresponding patient tumors to date. [...]"

The authors perform QC analysis on the current PDX biobank based on known spontaneous patient immune-cell derived grafts. Did the authors also check for spontaneous mouse tumours, especially for those with really fast growth rates? Spontaneous mouse tumours can arise at any given point in the family line of a given model. Proof of concept established

Also, some genotyping to confirm a match to the human originating sample would be useful.

Author response: *This is an excellent point, and one that we have encountered in another project we have generating PDXs of brain metastases. Pathological review and confirmation with species-specific PCR revealed a single spontaneous mouse tumor from a NOG mouse after >200-day latency period. Our WGS/RNA seq data confirms that all of the breast PDXs sequenced in this study are human in origin as they were aligned against human genomes (patient germline DNA or reference genome for RNA). While we did not perform WGS on multiple passages (due to costs), every PDX tumor, including*

passages and drug-treated mice, underwent histological analysis where there was no evidence of murine tumors.

Is there an association between P1/P2 growth rates and time to progression?

Author response: While there was a trend for patients who progressed within the timeframe of this study to have more rapid growth kinetics, there was not a statistically significant association between P2 growth rates and time to progression (see below; similar results for P1, data not shown). Therefore, this was not included in the manuscript.

In the discussion, this sentence “In other tumor types, the initial PDX maintained its predictive capacity even after late relapse and/or intervening lines of therapy (11)” would need to go linked to something like “needs further investigation”

Author response: We have modified this sentence to highlight the preliminary nature of this data which would require validation in another series: “Izumchenko et al. have demonstrated that PDXs from multiple tumor types accurately recapitulate the patient’s clinical response even after late relapse and/or intervening lines of therapy, though further studies prospectively demonstrating the predictive capacity of PDXs are warranted (11).”

Major comments

There is heterogeneity in drug responses amongst the PDXs tested, which seems to correlate with observations in the clinic. Although number model-drug tests are not high enough to make this conclusion, plus I don’t think is the scope nor the relevance of this manuscript (and has been done before by Gao et al. Nat Med 2015). Overall, PDXs seem to be responding less to Dox than to Pacli. The manuscript indeed shows PDXs do not respond to Dox, regardless of the response to the patient. The authors do mention this could be due to insufficient drug concentration used for the Dox trials due to mouse-specific toxicity. If so, other confounding factors related to drug PK/PD could be leading to misleading interpretations, consistent with the still high discordant results between in

vivo and clinical responses. I understand it is difficult to test the match drug response clinical data in patients. The authors should at least discuss this clearly in the discussion (including possible differences in metabolisms of the drug, differences in PK/PD affecting drug's blood and tumour concentrations, the difficulty of comparing to match single agent chemo treatment in patients, and the difference between "retaining" to "predicting" ...).

Author response: *We have incorporated these points in the Discussion as follows: "Our data is consistent with retention of chemosensitivity upon xenotransplantation, yet the prospect of using PDXs as avatars in a predictive setting to guide therapy will require co-clinical studies (eg. REFLECT study, NCT02732860). While cases representing clinical challenges were purposefully selected for inclusion in our study, chemogenomic profiling revealed nearly all models displayed actionable features that could be interrogated, several of which demonstrated marked regressions in vivo (Fig. 6). While the drugs used in our study have been approved in other indications and/or safely used in humans, evaluating experimental agents in PDXs must account for species differences in pharmacokinetics/dynamics so that they are tested using clinically-relevant dosing."*

The manuscript main observation in my opinion is that their data suggests heavily treated tumours are generally less sensitive to further lines of chemo treatment, and their data on concordance and discordant is weaker, also because logic follows that treatment will change their drug response profile. Do they have any match PDX treatment naïve vs post-treatment or pre and post-treated that could be tested? And hence highlight the relevance of such effect? Or could this be treatment specific?

Author response: *This question is best addressed with specimens from pre- and post-neoadjuvant chemotherapy, but unfortunately this study was not designed or REB-approved for additional pre-neoadjuvant biopsies. We are continuing to follow patients for whom we had developed treatment naïve PDXs who subsequently underwent adjuvant chemotherapy, so that we can attempt PDX development from their recurrent post-treated tumors, however, this will require several years of follow-up.*

How do you interpret VAF changes? $R < 0.3$ indicates a strong clonal drift... how is this consistent with previous literature? And is there anything that explains such heterogeneity in VAF dynamics upon xenotransplantation in these models?

Author response: *There are two other major studies which have performed WGS on breast PDXs, primary and germline DNA with following findings:*

Li et al (2013) evaluated 13 models with median $R = 0.71$ (range 0.26-0.85)

Eirew et al (2015) evaluated 13 models with median $R = 0.74$ (range 0.20-0.91)

In our study, we evaluated 34 models with median $R = 0.63$ (range -0.02-0.85)

These differences in VAF R values were not statistically significant (see below). It should also be noted that Eirew et al (2015) had not performed 2 VAF correlations due to low cellularity in the patient sample, while all tumors were included in our analysis. Low patient

tumor cellularity appears to contribute to low VAF correlations for models with the lowest R values in our series (see Supplementary Figure 5 – GCRC1882 and GCRC2001). While we cannot exclude clonal drift occurring in our models as has previously been shown to occur upon xenotransplantation by Eirew et al (2015), the functional implications of this remain unclear, given the strong preservation of driver mutations and transcriptional landscape. We have added this to the Discussion of our paper as follows: “This collection also represents the largest series of distinct breast cancer PDX models which have undergone WGS with their corresponding patient tumors. While the mutational load, copy number profiles and driver alterations were robustly conserved, the strength of VAF correlations was variable across PDX lines (Fig. 2, Supplementary Fig. 5). Eirew et al. have previously examined this using deep and single-cell sequencing to delineate multiple patterns of clonal dynamics upon xenotransplantation (15). Given that we and others have demonstrated preservation of the transcriptional and protein signaling outputs in PDXs, the functional significance of these subclonal shifts remain unclear (12–14,41).”

The “one mouse per treatment per model” approach is not the best approach to test drug responses in few models. However, I understand the financial and logistic implications of doing a 3x3 approach, plus the authors show when using the later, the results were very similar. A mention in the discussion that such 1x1x1 approach is better suited for large scale PDX preclinical trials which compensate the loss of measurement accuracy on individual mice would be good...

Author response: While the Gao et al. (2015) paper which popularized the 1x1x1 approach evaluated a large number of single-animal response curves for multiple agents across multiple tumor types, within a single tumor type they used 33-64 unique PDX lines, which is in line with the size of our cohort of breast PDXs. We have found value in another of ongoing work and other publications using a cohort of this size and have added the following to the Discussion: “While measurement accuracy is sacrificed in search for clinically meaningful tumor regressions using hundreds of unique models in the 1x1x1 approach, we have also found value in evaluating drug sensitivity across ~30 PDX models with substantial molecular heterogeneity for signal detection in discovery-phase experiments. By screening a portion of this PDX cohort for gefitinib sensitivity, we have

identified a subset of TNBCs which highly express EGFR specifically within the tumorigenic subpopulation of their tumors which renders them vulnerable to EGFR inhibition, as opposed to uniformly high EGFR expression which did not correlate with sensitivity (42)."

"This classification was consistent with clinical outcomes, where patients with chemoresistant PDXs displayed worse PFS... "... this is a circular argument. Chemo resistant PDXs are those from heavily treated patients which are likely to be at a more progressive state of the disease, and hence have a worse prognosis.

Author response: *We agree with the reviewer, however, the goal here is to demonstrate retention of aggressive biological features in the PDX. As with Fig. 1C, using experimental data from the PDX only (and no patient clinicopathologic features), we are able to stratify patients into PFS outcomes by ability to generate PDXs. In Fig. 5F, using PDX chemosensitivity data to classify patients (without patient clinicopathologic data) demonstrates that patients with tumors that generate chemoresistant PDXs have significantly worse PFS.*

The fidelity of the PDX response was assessed as a clinical cohort and from the perspective of an individual patient. At the population level—ref 23-25.. has the rate of response to single agents been evaluated in a clinical cohort? And is the clinical cohort similar to the preclinical one ? If not, this is an overstatement.

Author response: *These references are only single-agent studies in the metastatic setting. Unfortunately, many of these early studies did not evaluate ER/HER2 status. We have modified the sentence to: "The objective response rates (CR+PR) for each agent ranged from 8.0-51.5%, in line with those observed in single-agent studies in patients with metastatic breast cancers (Fig. 5B)(24–26)."*

"The mechanism by which three cases went from paclitaxel resistant in the patient to sensitive in the PDX remains unclear, though the effect of drug holiday during PDX generation resulting in re-sensitization may play a role". This has been observed in other labs. It could also be because engraftment is done in treatment naïve mice, allowing the highly proliferating cells killed by chemo to regrow and repopulate the bulk of the tumour.

Author response: *This is an excellent point. In addition to the clinical reference, we have added a reference for an excellent laboratory study on drug holiday/re-challenge: "The mechanism by which three cases went from paclitaxel resistant in the patient to sensitive in the PDX remains unclear, though the effect of drug holiday during PDX generation resulting in re-sensitization may play a role (28,29)."*

REVIEWERS' COMMENTS:

Reviewer #1 (Remarks to the Author):

The authors have addressed all of my concerns and I think the findings are very interesting and well presented.

Reviewer #2 (Remarks to the Author):

The authors have thoroughly addressed the points raised and improved the manuscript with their revisions.

Reviewer #3 (Remarks to the Author):

I am happy with the authors' rebuttal to my comments and their additions and changes in the manuscript.